

# Duality and hidden symmetry breaking in the q-deformed Affleck-Kennedy-Lieb-Tasaki model

Tyler Franke* and Thomas Quella[†]

The University of Melbourne, School of Mathematics and Statistics,
Parkville 3010 VIC, Australia

* Tyler.Franke@unimelb.edu.au , † Thomas.Quella@unimelb.edu.au

## Abstract

We revisit the question of string order and hidden symmetry breaking in the $q$-deformed AKLT model, an example of a spin chain that possesses generalized symmetry. We first argue that the non-local Kennedy-Tasaki duality transformation that was previously proposed to relate the string order to a local order parameter leads to a non-local Hamiltonian and thus does not provide a physically adequate description of the symmetry breaking. We then present a modified non-local transformation which is based on a recently developed generalization of Witten's Conjugation to frustration-free lattice models and capable of resolving this issue.

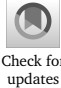

## 1   Introduction

The classification and characterization of phases of quantum matter and the associated phase transitions is one of the most profound challenges of theoretical and mathematical physics. Landau's paradigm has long provided a strong foundation for this task, resting on the idea of spontaneous symmetry breaking. The central ingredients of this theory are the symmetries of the Hamiltonian as well as their action on the space of ground states and order parameters associated with spontaneous symmetry breaking (SSB). Spontaneous symmetry breaking has been instrumental in the understanding of ferromagnetism, superconductivity and liquid crystals, to name just a few of the associated physical phenomena.

However, as knowledge about individual quantum systems advanced, it has become increasingly evident that Landau's paradigm in its historical form offers only a restricted view of the vast landscape of quantum phases of matter, specifically in the presence of exotic phases that exhibit robust topological properties such as topological insulators [1] or fractional quantum Hall systems [2]. While genuine topological phases require long-range entanglement, there also exist phases with short-range entanglement that cannot be removed if symmetries are enforced [3]. These phases are called symmetry-protected topological (SPT) phases. Recently, systematic attempts have been made to reconcile the existence of (symmetry-protected) topological phases with Landau's paradigm based on the consideration of generalized symmetries and their breaking, see [4,5] and references therein. These generalized symmetries come in different flavors and a truly general theory is still lacking but evidence suggests that (higher) categorical structures are a key ingredient (see, e.g., [6,7]).

The prototypical example of a non-trivial bosonic SPT phase is the AKLT model [8,9], a gapped one-dimensional isotropic spin chain of spin-1 degrees of freedom [10,11]. This model has a number of peculiar features that are intimately linked to its topological non-triviality: On a finite interval or a half-infinite chain it exhibits fractionalized spin-$\frac{1}{2}$ boundary spins [12,13]. The latter can also be inferred from the bipartite ground state entanglement spectrum of finite periodic or infinite chains [14] where the ground state is unique. Finally, the system displays non-local string order which measures diluted antiferromagnetism [15].

Soon after the emergence of the AKLT model, Kennedy and Tasaki related the existence of string order to what they coined "hidden symmetry breaking" in two influential papers [16,17]. More specifically, they considered the system with open boundary conditions and constructed a non-local transformation with the following key features:

1. It maps the original Hamiltonian to a new local Hamiltonian.

2. It preserves a $\mathbb{Z}_2 \times \mathbb{Z}_2$ subgroup of the spin rotation group $SO(3)$ that corresponds to rotations by $\pi$ about the three coordinate axes.[1]

3. The four ground states associated with the two fractionalized spin-$\frac{1}{2}$ boundary spins are mapped to product states and these form an orbit under the group $\mathbb{Z}_2 \times \mathbb{Z}_2$.

4. The non-local string order is mapped to a local correlation function.

---

[1]Recall that we are in a setting with spin-1 or, more generally, integer spins so that the $\mathbb{Z}_2$ subgroup of $SU(2)$ acts trivially and we are left with an action of $SO(3) = SU(2)/\mathbb{Z}_2$.

As a consequence, the original system is mapped to a new one which exhibits spontaneous $\mathbb{Z}_2 \times \mathbb{Z}_2$ symmetry-breaking. Shortly after, the results just sketched were revisited and generalized to higher (integer) spins by Oshikawa [18] (see also [19]). From a modern perspective, the Kennedy-Tasaki (KT) transformation establishes a duality between an SPT phase and a symmetry-broken phase of $\mathbb{Z}_2 \times \mathbb{Z}_2$ symmetric quantum systems. Generalizations of this duality and the hidden symmetry breaking picture to other symmetry groups have been discussed in [20,21] (see also [22] for a recent systematic discussion in the context of topological holography). The Kennedy-Tasaki transformation and generalizations thereof have since been a very active area of research [20,21,23–25]. In this context, we would like to highlight a recent definition and thorough discussion of the KT transformation for periodic boundary conditions in terms of a non-invertible defect [23].

In this paper we explore a complementary direction and revisit the $q$-deformation of the AKLT model which is invariant under the quantum group $U_q[\mathfrak{su}(2)]$ [26–28]. This spin chain can be thought of as an anisotropic and chiral version of the original AKLT model where spin rotation symmetry, time-reversal symmetry and inversion symmetry are broken. Despite this crucial difference, it has been known for a long time that many of its physical features carry over almost identically, such as the presence of fractionalized boundary spins or the presence of non-local string order [29]. More recently it was therefore suggested that the $q$AKLT model should be regarded as an example of an SPT phase [30,31], albeit with respect to the generalized symmetry $SO_q(3)$ which should be thought of as a global version of $U_q[\mathfrak{su}(2)]$. In these papers it was also suggested that discrete duality-type symmetries might already be sufficient to protect the topological properties, in close analogy to what is known for the undeformed case.

Let us explain this point in more detail. One curious feature of the $q$AKLT model is the explicit breaking of not only the continuous rotation symmetry but also almost all standard discrete group-symmetries of the AKLT model, especially those symmetries that are known to protect the AKLT model's topological features (see [8]). This includes specifically time-reversal, inversion and the $\mathbb{Z}_2 \times \mathbb{Z}_2$ subgroup of $\pi$-rotations about the principal axes. In the $q$-deformed case, most of these transformations only remain symmetries if they are accompanied by a $q \to q^{-1}$ duality transformation [29–31]. In the case of the rotations, the resulting transformations constitute the duality group $\mathbb{Z}_2 \times \mathbb{Z}_2^{(q)}$, where the superscript indicates the non-trivial action on the parameter $q$.

Despite these substantial differences, it was argued in [29] that the standard Kennedy-Tasaki transformation maps the non-local string-order present in the $q$AKLT model to local order capable of detecting a $\mathbb{Z}_2 \times \mathbb{Z}_2^{(q)}$ symmetry breaking.[2] However, as will be explained in more detail in Section 2.2 below, the transformation proposed in [29] has one serious flaw which hitherto seemingly went unnoticed: The transformed Hamiltonian ceases to be local, thus questioning the physical interpretation as spontaneous symmetry breaking.[3]

The goal of this paper is to describe a modified duality transformation that corrects this issue. Our construction is based on a specific implementation of Witten's Conjugation, a tool for the systematic deformation of frustration-free Hamiltonians that preserves the interaction range [32]. As one concrete application, the aforementioned paper showed that the $q$-deformation of the AKLT model can be implemented by means of Witten's Conjugation [32]. By composing Witten's Conjugation with the standard Kennedy-Tasaki transformation we are able to construct a non-local transformation that implements hidden symmetry breaking while preserving locality of the Hamiltonian. The validity of our approach is confirmed with an in-depth discussion of the associated symmetry-broken quantum states. We posit that the Hamiltonian

---

[2]The authors of [29] focused on the $\mathbb{Z}_2$ factor and spoke of a "partial" symmetry breaking. However, we find it more appropriate to use different language, see Section 4.2 for a more detailed discussion.

[3]We thank Robert Pryor for sharing that observation and discussions on that point.

we construct is a representative of the $\mathbb{Z}_2 \times \mathbb{Z}_2^{(q)}$ spontaneous symmetry breaking phase.

This paper is organized as follows. In Section 2 we review the standard hidden symmetry picture in detail and describe in what sense it fails after $q$-deformation. The idea of Witten's Conjugation and its specific application to the AKLT model is described in Section 3. All ingredients are then combined in Section 4 which features the construction and physical interpretation of our new duality transformation. The Conclusions summarize some open problems and future directions. Finally, there are some appendices with more technical material. This includes an explanation of why $\mathbb{Z}_2 \times \mathbb{Z}_2^{(q)}$ symmetry is a natural consequence of the quantum group symmetry in the context of general $SO(3)_q$-symmetric spin chains.

## 2 The Kennedy-Tasaki transformation and hidden symmetry breaking

In Section 2.1 we review the Kennedy-Tasaki duality [17, 18], which is the prototype of hidden symmetry breaking. We then show in Section 2.2 that this duality transformation leads to non-local interactions when applied to the $q$AKLT Hamiltonian for non-trivial $q$.

### 2.1 Hidden $\mathbb{Z}_2 \times \mathbb{Z}_2$ symmetry breaking in the AKLT model

The prototype of SPT ordered spin chains is the AKLT model [10, 11]. The AKLT Hamiltonian can be written as a sum over projectors acting on nearest-neighbors. Such Hamiltonians are frustration-free, i.e. the ground states can be obtained by minimizing each contribution separately. As a consequence, the ground states of the AKLT model can be obtained exactly. In terms of SU(2) spin operators $\vec{S}_k = (S_k^x, S_k^y, S_k^z)$, the spin-1 AKLT Hamiltonian for an open chain of length $L$ is written

$$H_{\text{AKLT}} = \sum_{j=1}^{L-1} \left\{ \frac{1}{6} \left( \vec{S}_j \cdot \vec{S}_{j+1} \right)^2 + \frac{1}{2} \vec{S}_j \cdot \vec{S}_{j+1} + \frac{1}{3} \right\}. \tag{1}$$

This Hamiltonian has $SO(3)$ spin-rotation symmetry. When enforcing periodic boundary conditions the AKLT Hamiltonian has a unique ground state which can be cast in the form of a matrix product state (MPS) (see [33] for a review and [34] for the earlier notion of finitely correlated states),

$$|\text{AKLT}\rangle = \text{tr}(A_1 A_2 \cdots A_L), \quad \text{with} \quad A_j = \sqrt{\frac{2}{3}} \begin{pmatrix} -\frac{1}{\sqrt{2}}|0\rangle_j & |+\rangle_j \\ -|-\rangle_j & \frac{1}{\sqrt{2}}|0\rangle_j \end{pmatrix}, \tag{2}$$

where $\{|0\rangle, |\pm\rangle\}$ are the standard spin-1 basis states labeled by their $S^z$ eigenvalue. In contrast, for a system with open boundary conditions every matrix element of the product $A_1 \cdots A_L$ corresponds to an independent ground state, thus yielding a four-fold degeneracy. From a physical perspective, this degeneracy is associated with the presence of spin-$\frac{1}{2}$ edge modes. We will use $|\text{AKLT}\rangle_{\alpha\beta} = (A_1 \cdots A_L)_{\alpha\beta}$ to denote the AKLT state with left and right boundary spin configurations $S^z = \alpha$ and $S^z = \beta$ respectively.

The MPS in Equation (2) can also be written

$$|\text{AKLT}\rangle = \sum_{\{\sigma_i\}=0,\pm} \text{tr}(A_1^{\sigma_1} A_2^{\sigma_2} \cdots A_L^{\sigma_L})|\sigma_1 \sigma_2 \cdots \sigma_L\rangle, \tag{3}$$

where $A_j^{\sigma_j}$ is the $\sigma_j$ component of the MPS tensor. That is,

$$A_j^0 = \sqrt{\frac{1}{3}}\begin{pmatrix} -1 & 0 \\ 0 & 1 \end{pmatrix}, \qquad A_j^+ = \sqrt{\frac{2}{3}}\begin{pmatrix} 0 & 1 \\ 0 & 0 \end{pmatrix}, \qquad A_j^- = \sqrt{\frac{2}{3}}\begin{pmatrix} 0 & 0 \\ -1 & 0 \end{pmatrix}. \tag{4}$$

Each matrix element in the products $A_1^{\sigma_1} A_2^{\sigma_2} \cdots A_L^{\sigma_L}$ now defines a product state present in the MPS with a given edge mode configuration. The form of the MPS tensor forbids the appearance of consecutive $S^z = \pm$ configurations in the AKLT ground states:

$$A_j^\pm A_{j+1}^\pm = 0. \tag{5}$$

Since $A^0$ is diagonal, the above relations similarly imply $A_i^\pm A_{i+1}^0 \cdots A_j^0 A_{j+1}^\pm = 0$. Contributions to the ground states then possess diluted antiferromagnetic order in the sense that the magnetic quantum numbers need to alternate if one ignores sites on which the local spin satisfies $S^z = 0$ [15,17]. As the number and length of such "defects" can be arbitrary, diluted antiferromagnetic order cannot be measured by a local order parameter. This is also evident from the fact that two-point correlation functions, which form the order parameter for antiferromagnetic order, decay exponentially in the AKLT ground state [11].

A string can be inserted into the two-point correlation functions to account for these defects, leading to the string order parameter [15][4]

$$\mathcal{O}_{\mathrm{SO}}^\alpha(\psi) = \lim_{|i-j|\to\infty} \langle\psi|S_i^\alpha \prod_{k=i+1}^{j-1} e^{i\pi S_k^\alpha} S_j^\alpha|\psi\rangle. \tag{6}$$

The string multiplies the correlation function $S_i^\alpha S_j^\alpha$ by a factor $(-1)^n$, where $n$ is the number of spin flips between sites $i$ and $j$. This serves the same role as the factor $(-1)^{|i-j|}$ in the Néel order parameter

$$\mathcal{O}_{\mathrm{Néel}}^\alpha(\psi) = \lim_{|i-j|\to\infty} (-1)^{|i-j|}\langle\psi|S_i^\alpha S_j^\alpha|\psi\rangle, \tag{7}$$

which ensures $(-1)^{|i-j|}S_i^\alpha S_j^\alpha = 1$ in an antiferromagnetically ordered state.

The string order parameters are indeed non-vanishing in the AKLT state, with $\mathcal{O}_{\mathrm{SO}}^\alpha(\mathrm{AKLT}) = -4/9$ for $\alpha = x$, $y$ or $z$ [16, 17]. This is true for periodic as well as open boundary conditions and regardless of the choice of boundary spins.

The order detected by the string order parameter is associated to symmetry breaking via the Kennedy-Tasaki (KT) duality. Having originally been developed as a duality for the spin-1 AKLT model [16, 17], the KT duality was then generalized to apply to general integer spin chains with $\mathbb{Z}_2 \times \mathbb{Z}_2$ symmetry [18]. The duality is implemented on integer spin chains with open boundary conditions by the non-local *unitary* operator [18]

$$U_{\mathrm{KT}} = \prod_{j=1}^{L} \exp\left[i\pi \sum_{k=1}^{j-1} S_k^z S_j^x\right]. \tag{8}$$

This operator is invariant under $\mathbb{Z}_2 \times \mathbb{Z}_2$, generated by $\pi$-(spin )rotations about the principal axes, which we denote with $R^\alpha = e^{i\pi S^\alpha}$, where $\alpha = x$, $y$ or $z$, but not the full group $SO(3)$ generated by spin rotations. Thus $SO(3)$-invariant operators will generally have their symmetry reduced to $\mathbb{Z}_2 \times \mathbb{Z}_2$ under conjugation by $U_{\mathrm{KT}}$. This is especially true for the operator,

$$H_{\mathrm{SSB}} = U_{\mathrm{KT}} H_{\mathrm{AKLT}} U_{\mathrm{KT}}^{-1}, \tag{9}$$

---

[4]Left implicit in (6) is the thermodynamic limit $L \to \infty$. In any expression where we take the distance between sites to be infinite the thermodynamic limit is assumed.

that arises as the image of the AKLT Hamiltonian. We call the resulting operator $H_{\text{SSB}}$, in anticipation of the fact that it is the Hamiltonian associated with a phase exhibiting full $\mathbb{Z}_2 \times \mathbb{Z}_2$ spontaneous symmetry breaking (SSB). From a general perspective, $H_{q\text{AKLT}}$ and $H_{\text{SSB}}$ should both have finite-range interactions in order to interpret the transformation as a duality, see Section 3.1 in [35]. $H_{\text{SSB}}$ is indeed also a nearest-neighbor Hamiltonian [16, 17].

As this point is crucial for our discussion of the $q$-deformed case, let us provide a more explicit description of the KT transformation and its action on various operators. Under conjugation by $U_{\text{KT}}$, local spin operators $\vec{S}_k$ will be mapped to string operators, and since $U_{\text{KT}}$ is an involution the converse is also true. This is summarised by the relations [18]

$$
\begin{aligned}
U_{\text{KT}} S_j^x U_{\text{KT}}^{-1} &= S_j^x \prod_{k=j+1}^{L} e^{i\pi S_k^x}, \\
U_{\text{KT}} S_j^y U_{\text{KT}}^{-1} &= \prod_{k=1}^{j-1} e^{i\pi S_k^z} S_j^y \prod_{l=j+1}^{L} e^{i\pi S_l^x}, \\
U_{\text{KT}} S_j^z U_{\text{KT}}^{-1} &= \prod_{k=1}^{j-1} e^{i\pi S_k^z} S_j^z.
\end{aligned}
\tag{10}
$$

In order to find an interaction that is mapped to a local operator, the strings need to (mostly) cancel out in pairs. This is the case for terms of the form $S_j^\alpha S_{j+1}^\alpha$, $(S_j^\alpha)^2$ and expressions containing each of the three spin generators precisely once (e.g. $S_j^x S_{j+1}^y S_j^z$), as well as polynomials in these building blocks. We stress that these are all of the possible $\mathbb{Z}_2 \times \mathbb{Z}_2$-invariant terms that can appear in a nearest-neighbor Hamiltonian which is a polynomial in the spin operators. In other words, the KT transformation maps such a nearest-neighbor Hamiltonian to a nearest-neighbor Hamiltonian only if it is $\mathbb{Z}_2 \times \mathbb{Z}_2$-invariant. This is a consequence of the more general fact that for a symmetry $\mathcal{S}$, duality transformations between $\mathcal{S}$-invariant Hamiltonians preserve the support of only $\mathcal{S}$-invariant operators under conjugation [22].

The relations (10) also imply two-point correlation functions and string correlation functions are mapped to each other under conjugation (we assume $i < j$),

$$
\begin{aligned}
U_{\text{KT}} S_i^x S_j^x U_{\text{KT}}^{-1} &= S_i^x \prod_{k=i+1}^{j} e^{i\pi S_k^x} S_j^x, \\
U_{\text{KT}} S_i^y S_j^y U_{\text{KT}}^{-1} &= e^{i\pi S_i^z} S_i^y \prod_{k=i+1}^{j-1} e^{i\pi S_k^y} e^{i\pi S_j^x} S_j^y, \\
U_{\text{KT}} S_i^z S_j^z U_{\text{KT}}^{-1} &= S_i^z \prod_{k=i}^{j-1} e^{i\pi S_k^z} S_j^z.
\end{aligned}
\tag{11}
$$

Thus the four AKLT ground states are each related to SSB states via the KT duality, in that $U_{\text{KT}}$ maps these states to states in which two point correlation functions do not decay. We then define $|\text{SSB}\rangle_{\alpha\beta} \equiv U_{\text{KT}} |\text{AKLT}\rangle_{\alpha\beta}$ to be the SSB states corresponding to each possible edge mode configuration in the AKLT state. Taking the $z$-axis string order parameter measured in the spin-1 AKLT state for example, the relations (11) imply [18]

$$
\begin{aligned}
\mathcal{O}_{\text{SO}}^z(\text{AKLT}) &= \lim_{|i-j|\to\infty} \langle \text{AKLT}| S_i^z \prod_{k=i+1}^{j-1} e^{i\pi S_k^z} S_j^z |\text{AKLT}\rangle \\
&= -\lim_{|i-j|\to\infty} \langle \text{SSB}| S_i^z S_j^z |\text{SSB}\rangle,
\end{aligned}
\tag{12}
$$

regardless of the choice of boundary condition. Similar relations hold between string and two-point correlation functions for the $x$- and $y$-axis spin operators.

Applying the KT transformation to the spin-1 AKLT Hamiltonian, one obtains a new $\mathbb{Z}_2 \times \mathbb{Z}_2$-invariant Hamiltonian $H_{\text{SSB}} \equiv U_{\text{KT}} H_{\text{AKLT}} U_{\text{KT}}^{-1}$ that is again a sum over local projectors and hence frustration-free [12, 17]. The space of ground states of the KT dual model $H_{\text{SSB}}$ is spanned by four product states [16, 17],[5]

$$|\Phi_{1,\pm}\rangle = |\phi_{1,\pm}\rangle^{\otimes L}, \quad \text{and} \quad |\Phi_{2,\pm}\rangle = |\phi_{2,\pm}\rangle^{\otimes L}, \tag{13}$$

where

$$|\phi_{1,\pm}\rangle = \frac{1}{\sqrt{3}}\left(|0\rangle + \sqrt{2}|\pm\rangle\right), \quad \text{and} \quad |\phi_{2,\pm}\rangle = \frac{1}{\sqrt{3}}\left(|0\rangle - \sqrt{2}|\pm\rangle\right). \tag{14}$$

In this form we see that each state is ferromagnetically ordered. Indeed, the local magnetization is non-zero in the thermodynamic limit when measured in any of these states. In agreement with relation (12) the two-point correlation functions are non-vanishing in these states. The action of $\mathbb{Z}_2 \times \mathbb{Z}_2$ permutes between ground states in the following way: $R^z$ exchanges the labels 1 and 2 inside the ket, $R^x$ the labels $\pm$, and $R^y$ both labels.

The basis (14) of ferromagnetically ordered states can be equivalently obtained by applying the KT transformation to the AKLT state for each set of edge spin configurations. The ground state degeneracy in the open AKLT model is then associated to spontaneous symmetry breaking, corresponding to bulk ferromagnetic order, in the KT dual model. Thus the symmetry breaking in the AKLT model is said to be hidden, and revealed via the KT duality.

## 2.2 Breakdown of hidden $\mathbb{Z}_2 \times \mathbb{Z}_2$ symmetry breaking for non-trivial $q$

An anisotropic version of the AKLT model can be obtained by considering a $q$-deformation of the $SO(3)$ symmetry [26–28]. This is done by replacing the symmetry algebra $\mathfrak{su}(2)$ with the quantum group $U_q[\mathfrak{su}(2)]$.[6] The local $q$AKLT Hamiltonian is still defined as a sum over projectors, though now onto $U_q[\mathfrak{su}(2)]$ spin states.[7] Thus when enforcing suitable generalized periodic boundary conditions, the $q$AKLT model has a unique ground state given by an MPS, which is derived in Appendix C,[8]

$$|q\text{AKLT}\rangle = \text{tr}\left(q^{2S^z} A_1(q) \cdots A_L(q)\right), \quad \text{with} \quad A_j(q) = \Omega \begin{pmatrix} -\frac{q^{-1}}{\sqrt{q+q^{-1}}}|0\rangle_j & q^{\frac{1}{2}}|+\rangle_j \\ -q^{-\frac{1}{2}}|-\rangle_j & \frac{q}{\sqrt{q+q^{-1}}}|0\rangle_j \end{pmatrix}, \tag{15}$$

where $\Omega = \sqrt{(q+q^{-1})/(q^2 + q^{-2} + 1)}$. Because the bond dimension of the MPS tensor is two, there is a fourfold ground state degeneracy for the open chain and exponentially decaying two-point correlation functions, as in the AKLT model.

Note that Equation (5) also holds for the $q$AKLT MPS tensor. There is then diluted antiferromagnetic order in the $q$AKLT state. This order is detected by a string order parameter, namely we have $\mathcal{O}_{\text{SO}}^z(q\text{AKLT}) \neq 0$ [27, 29]. In contrast, the string order parameters associated to $x$- and $y$-axis spin operators vanish in the $q$AKLT state for $q \neq 1$. Just as in the undeformed case, these three string order parameters are mapped to local correlation functions using the KT transformation [29]. For $q \neq 1$ only one of the three is non-zero, inspiring the authors of Ref. [29] to use the term "partial" hidden symmetry breaking to refer to this phenomenon.

---

[5]These states and the dual Hamiltonian $H_{\text{SSB}}$ are discussed in greater detail in Section 4.

[6]See Appendix A for a brief summary of properties of $U_q[\mathfrak{su}(2)]$ and its representations.

[7]This construction is done explicitly in Section 3.2.

[8]The operator $q^{2S^z}$ is inserted into the trace to enforce boundary conditions which preserve the $U_q[\mathfrak{su}(2)]$ symmetry in the Hamiltonian, the importance of which was highlighted in Ref. [30].

In the current paper we will adopt a slightly different use of language which, in our opinion, provides a more accurate description of the physical situation. The key point is that the factor $R^z$ in $\mathbb{Z}_2 \times \mathbb{Z}_2$ is *explicitly* broken by the $q$AKLT Hamiltonian and hence also by its image under the KT transformation. In this sense, the symmetry breaking described by the order parameters is full, not partial, with respect to the remaining symmetry $\mathbb{Z}_2 = \langle \mathbb{1}, R^z \rangle$.

However, one can even go one step further: We will argue that it is meaningful to view the full symmetry as $\mathbb{Z}_2 \times \mathbb{Z}_2^{(q)}$ which also includes the duality transformations. And with respect to this group of duality symmetries one actually still encounters *full* symmetry breaking, in analogy to the undeformed case.

However, before we return to this point we would like to address a subtlety that is directly linked to the physical interpretation. Whether or not the $q$AKLT Hamiltonian transforms to a local Hamiltonian upon applying the KT transformation was not investigated in [29]. We show here that the $q$AKLT Hamiltonian becomes a non-local Hamiltonian under the KT transformation whenever $q \neq 1$. This observation spoils the interpretation of the KT transformation as a map from the Haldane phase to a symmetry-broken phase for generic choices of $q$.

The non-locality can be traced back to there not being a $\mathbb{Z}_2 \times \mathbb{Z}_2$ symmetry generated by $R^x$, $R^y$ and $R^z$ in the $q$AKLT model. Recall from the discussion around (10) that a local Hamiltonian will remain local after conjugation by the KT transformation only if it is $\mathbb{Z}_2 \times \mathbb{Z}_2$ invariant. Applying the KT transformation to the local $q$AKLT Hamiltonian then must result in a non-local Hamiltonian, since it is not $\mathbb{Z}_2 \times \mathbb{Z}_2$ invariant. In what follows we will see that non-local terms are attached to both bulk and boundary in the transformed Hamiltonian.

Let us explain these points in more detail. In terms of the standard $SU(2)$ spin operators, the spin-1 $q$AKLT Hamiltonian can be written [26, 28]

$$
\begin{aligned}
H_{q\text{AKLT}} = b \sum_{j=1}^{L-1} \Bigg\{ & c\mathbf{S}_j \cdot \mathbf{S}_{j+1} + \Big[\mathbf{S}_j \cdot \mathbf{S}_{j+1} + \frac{1}{2}(1-c)(q+q^{-1}-2)S_j^z S_{j+1}^z \\
& + \frac{1}{4}(1+c)(q-q^{-1})(S_{j+1}^z - S_j^z)\Big]^2 + \frac{1}{4}c(1-c)(q+q^{-1}-2)^2(S_j^z S_{j+1}^z)^2 \\
& + \frac{1}{4}c(1+c)(q+q^{-1})(q+q^{-1}-2)S_j^z S_{j+1}^z(S_{j+1}^z - S_j^z) \\
& + \frac{1}{4}(c-3)\Big[(c-1+\frac{1}{2}(1+c)^2 S_j^z S_{j+1}^z + 2\big(c-\frac{1}{8}(1+c)^2\big)\big((S_{j+1}^z)^2 + (S_j^z)^2\big)\Big] \\
& + (c-1) + \frac{1}{2}c(q^2-q^{-2})(S_{j+1}^z - S_j^z)\Bigg\},
\end{aligned}
\tag{16}
$$

with $c = 1 + q^2 + q^{-2}$ and $b = \big[c(c-1)\big]^{-1}$. There are two terms in the local $q$AKLT Hamiltonian that are not $\mathbb{Z}_2 \times \mathbb{Z}_2$-invariant:[9]

$$
S_j^z S_{j+1}^z(S_{j+1}^z - S_j^z), \quad \text{and} \quad S_{j+1}^z - S_j^z. \tag{17}
$$

As a consequence, the KT transformation creates non-local terms, specifically those of the form

$$
U_{\text{KT}} S_j^z S_{j+1}^z(S_{j+1}^z - S_j^z) U_{\text{KT}}^{-1} = -\exp\Big[i\pi \sum_{k=1}^{j-1} S_k^z\Big] S_j^z S_{j+1}^z\big(\exp\big[i\pi S_j^z\big]S_{j+1}^z - S_j^z\big). \tag{18}
$$

The duality picture is then spoiled since the proposed dual Hamiltonian is non-local.

---

[9]We note that the single-site operators cancel out in the bulk due to the summation over $j$. However, in semi-infinite or finite open system they will lead to boundary terms that are essential for the preservation of $U_q[\mathfrak{su}(2)]$ invariance.

Beyond the clear physical discrepancies, the non-locality means that any attempt at a correspondence fails to fit into systematic approaches to constructing dualities such as bond algebra isomorphisms [35,36], which have been studied using matrix product operator intertwiners [37,38] and topological holography [22,39]. Note that in the approaches just referred to duality transformations are defined with respect to the global symmetry of the model, and the existence of that symmetry is crucial to maintain locality in the dual model. This reinforces the argument around Equation (17) that the standard KT transformation is the incorrect transformation to use because the $q$AKLT model does not have $\mathbb{Z}_2 \times \mathbb{Z}_2$ symmetry.

We now discuss the generalized symmetry which replaces $\mathbb{Z}_2 \times \mathbb{Z}_2$ in the $q$AKLT model. There is still a $\mathbb{Z}_2$ symmetry generated by $R^z$ present. Moreover, the $R^x$ and $R^y$ still leave the Hamiltonian invariant if one replaces $q$ by $q^{-1}$ [29,30]; they become duality transformations between the $q$AKLT and the $q^{-1}$AKLT model. The symmetry group $\mathbb{Z}_2 \times \mathbb{Z}_2$ is then replaced with what we call $\mathbb{Z}_2 \times \mathbb{Z}_2^{(q)}$ *duality-symmetry*: a combination of $\mathbb{Z}_2$ symmetry generated by $R^z$ and discrete dualities implemented by $R^x$ and $R^y$. Note that the map $q \to q^{-1}$ itself is not implemented via a linear transformation on the Hilbert space. Distinguishing between sending $q \to q^{-1}$ and mapping between the models with parameter $q$ and $q^{-1}$ with $\pi$-rotations about the principal axes is important in interpreting the patterns of spontanous symmetry breaking presented in Section 4.2. Any $U_q[\mathfrak{su}(2)]$-symmetric Hamiltonian automatically exhibits $\mathbb{Z}_2 \times \mathbb{Z}_2^{(q)}$ symmetry because it is a property of the corresponding quadratic Casimir element. A more detailed discussion on this final point can be found in Appendix B.

In what follows we provide a partial answer to a question posed by one of the current authors in [30]: What role does $\mathbb{Z}_2 \times \mathbb{Z}_2^{(q)}$ play in protecting the topological properties of the $q$AKLT model? To achieve this, we construct a Hamiltonian that lies in a $\mathbb{Z}_2 \times \mathbb{Z}_2^{(q)}$ SSB phase which can be considered dual to the $\mathbb{Z}_2 \times \mathbb{Z}_2^{(q)}$ SPT phase that the $q$AKLT Hamiltonian lies in. In particular, the edge states of the $q$AKLT Hamiltonian are mapped to product states that are ferromagnetically ordered. In our construction $\mathbb{Z}_2 \times \mathbb{Z}_2^{(q)}$ plays the same role as $\mathbb{Z}_2 \times \mathbb{Z}_2$ in the original KT duality. Our tool of choice is Witten's Conjugation which will be explained next.

## 3 Witten's conjugation

In Ref. [32] the authors adapted Witten's Conjugation argument for supersymmetric quantum systems, given in Ref. [40], to frustration-free spin chains. They used their result to connect existing frustration-free spin Hamiltonians as well as generate new ones. In particular, they were able to connect the AKLT and $q$AKLT Hamiltonians. In this section we outline the general argument given in Ref. [32] and review their example of the $q$AKLT Hamiltonian.

### 3.1 Connecting frustration-free spin Hamiltonians

Consider a 1-dimensional frustration-free nearest-neighbor Hamiltonian with $L$ spin sites and open boundary conditions,

$$H = \sum_{j=1}^{L-1} H_{j,j+1}, \quad \text{with} \quad H_{j,j+1} \geq 0. \tag{19}$$

By frustration-free we mean that a ground state $|\psi\rangle$ minimizes each local Hamiltonian,

$$H_{j,j+1}|\psi\rangle = 0, \quad \text{for all} \quad j \in \{1, 2, \cdots, L-1\}. \tag{20}$$

Say $\{|\psi_i\rangle\}_{i=1}^n$ is a basis for the space of ground states of $H$, that is $\text{span}_{\mathbb{C}}\{|\psi_i\rangle\}_{i=1}^n = \ker(H_{j,j+1})$ for all $j$. The hermiticity and positivity of the $H_{j,j+1}$ means that it can be written

$H_{j,j+1} = L^{\dagger}_{j,j+1} L_{j,j+1}$, where $L_{j,j+1}$ is a two-site operator. Note that this implies $\text{span}_{\mathbb{C}}\{|\psi_i\rangle\}^n_{i=1} = \ker(L_{j,j+1})$. Consider then the operator $M = \prod^L_{j=1} M_j$, where each $M_j$ is invertible and acts non-trivially only on site $j$. We call $M$ and $M_j$ the *Witten's Conjugation operator* interchangeably. Note that conjugating an operator with $M$ will preserve the support of that operator. Using the Witten's Conjugation operator we can then obtain a deformed local Hamiltonian,

$$\tilde{H} = \sum^{L-1}_{j=1} \tilde{H}_{j,j+1}, \quad \text{with} \quad \tilde{H}_{j,j+1} = \tilde{L}^{\dagger}_{j,j+1} C_{j,j+1} \tilde{L}_{j,j+1} \quad \text{and}, \quad \tilde{L}_{j,j+1} = M L_{j,j+1} M^{-1}, \quad (21)$$

where $C_{j,j+1}$ are positive definite Hermitian operators with support on sites $j$ and $j+1$.

The authors in [32] showed that the ground states of the deformed Hamiltonian are given by $\text{span}_{\mathbb{C}}\{M|\psi_i\rangle\}^n_{i=1} = \ker(\tilde{L}_{j,j+1})$. We repeat their argument here for completeness. It is immediate from the implementation of the deformation that $\tilde{H}_{j,j+1}$ is positive semi-definite and $\text{span}_{\mathbb{C}}\{M|\psi_i\rangle\}^n_{i=1} \subseteq \ker(\tilde{L}_{j,j+1})$. To show $\ker(\tilde{L}_{j,j+1}) \subseteq \text{span}_{\mathbb{C}}\{M|\psi_i\rangle\}^n_{i=1}$, consider an arbitrary state $|\phi\rangle \in \ker(\tilde{L}_{j,j+1})$. Then $M L_{j,j+1} M^{-1}|\phi\rangle = 0$, for any $j$. As $\text{span}_{\mathbb{C}}\{|\psi_i\rangle\}^n_{i=1} = \ker(L_{j,j+1})$ and the inverse for $M$ is unique, this condition is satisfied if and only if $|\phi\rangle = \sum^n_{i=1} a_i M|\psi_i\rangle$, where $a_i \in \mathbb{C}$ are constants. Thus $\ker(\tilde{L}_{j,j+1}) \subseteq \text{span}_{\mathbb{C}}\{M|\psi_i\rangle\}^n_{i=1}$, meaning $\ker(\tilde{L}_{j,j+1}) = \text{span}_{\mathbb{C}}\{M|\psi_i\rangle\}^n_{i=1}$ as desired.

Let us summarize briefly what we obtain from Witten's Conjugation. Starting with a local frustration-free Hamiltonian $H$ and a basis for the space of its ground states $\{|\psi_i\rangle\}^n_{i=1}$, we can obtain a new local frustration-free Hamiltonian $\tilde{H}$ if we are given an invertible operator $M = \prod^L_{j=1} M_j$ via the process in Equation (21). The states $\{M|\psi_i\rangle\}^n_{i=1}$ will form a basis of the space of ground states of $\tilde{H}$.

## 3.2 Mapping between the AKLT models

Following [32], we next review the application of Witten's Conjugation to map between the AKLT and $q$AKLT models for generic $q > 0$. As a preparation we write the $q$AKLT Hamiltonian (16) in the form

$$H_{q\text{AKLT}} = \sum^{L-1}_{j=1} H^{q\text{AKLT}}_{j,j+1}, \quad (22)$$

where the two-site Hamiltonians $H^{q\text{AKLT}}_{j,j+1}$ project onto the $U_q[\mathfrak{su}(2)]$-invariant spin-2 subspace in the tensor product $1 \otimes 1 = 0 \oplus 1 \oplus 2$. As opposed to the expression in terms of spin operators (16), it is more convenient for our purposes to express the local Hamiltonian in terms of the spin-2 basis states,

$$H^{q\text{AKLT}}_{j,j+1} = \sum^2_{m=-2} |\psi^{(q)}_m\rangle\langle\psi^{(q)}_m|. \quad (23)$$

We expand the spin-2 basis states in the basis states of the tensor product of two $U_q[\mathfrak{su}(2)]$ spin-1 representations,

$$\begin{aligned}
|\psi^{(q)}_{\pm 2}\rangle &= |\pm\pm\rangle, \\
|\psi^{(q)}_{\pm 1}\rangle &= \frac{1}{\sqrt{q^2+q^{-2}}}\Big[q^{\mp 1}|\pm 0\rangle + q^{\pm 1}|0\pm\rangle\Big], \\
|\psi^{(q)}_0\rangle &= \frac{1}{\sqrt{q^4+q^{-4}+(q+q^{-1})^2}}\Big[(q+q^{-1})|00\rangle + q^{-2}|+-\rangle + q^2|-+\rangle\Big].
\end{aligned} \quad (24)$$

This follows from the known Clebsch-Gordan coefficients or by explicit computation using the information in Appendix A.

The ground states of the $q$AKLT Hamiltonian with open boundary conditions can be written in MPS form,

$$|q\text{AKLT}\rangle_{\alpha\beta} = (A_1(q)A_2(q)\cdots A_L(q))_{\alpha\beta}, \tag{25}$$

where $\alpha$ and $\beta$ denote the choice of spin-$\frac{1}{2}$ edge modes and $A_j(q)$ is the $q$AKLT *MPS tensor* defined in Equation (15) (see Appendix C for a derivation). Setting $q = 1$ in Equations (16) and (25) we recover the AKLT Hamiltonian and the AKLT MPS tensor, respectively.

Now we explain how one can obtain the $q$AKLT Hamiltonian and MPS tensor starting from the undeformed case $q = 1$. First we discuss that the Witten's Conjugation operator $M$ based on the choices,

$$M_j = q^{-2jS_j^z}\left(\frac{q+q^{-1}}{2}\right)^{\frac{1}{2}(S_j^z)^2}, \tag{26}$$

takes us from the AKLT ground to the $q$AKLT ground state [32] (up to normalization and a gauge transformation). We then show that we also obtain the $q$AKLT Hamiltonian using the same operator.

Acting on the physical space of the AKLT MPS tensor with the Witten's Conjugation operator, we obtain[10]

$$M_j \triangleright A_j(q=1) = \sqrt{\frac{q+q^{-1}}{3}}\begin{pmatrix} -\frac{1}{\sqrt{q+q^{-1}}}|0\rangle_j & q^{-2j}|+\rangle_j \\ -q^{2j}|-\rangle_j & \frac{1}{\sqrt{q+q^{-1}}}|0\rangle_j \end{pmatrix}. \tag{27}$$

The gauge transformation $G_j = q^{(2j-1)S^z+\frac{1}{2}}$ removes the site-dependence in the off-diagonal terms,

$$\bar{A}_j(q) = G_j\big(M_j \triangleright A_j(q=1)\big)G_{j+1}^{-1} = \sqrt{\frac{q+q^{-1}}{3}}\begin{pmatrix} -\frac{q^{-1}}{\sqrt{q+q^{-1}}}|0\rangle_j & |+\rangle_j \\ -|-\rangle_j & \frac{q}{\sqrt{q+q^{-1}}}|0\rangle_j \end{pmatrix}. \tag{28}$$

Note that this gauge transformation will result in a non-trivial action on the boundaries when applied to the MPS (25).

The MPS tensor given in Equation (28) can be shown to be equivalent to the $q$AKLT MPS tensor. To see this, first note that

$$q^{\frac{S^z}{2}} \triangleright \bar{A}_j(q) = \sqrt{\frac{q+q^{-1}}{3}}\begin{pmatrix} -\frac{q^{-1}}{\sqrt{q+q^{-1}}}|0\rangle_j & q^{\frac{1}{2}}|+\rangle_j \\ -q^{-\frac{1}{2}}|-\rangle_j & \frac{q}{\sqrt{q+q^{-1}}}|0\rangle_j \end{pmatrix} = \sqrt{\frac{1+q^2+q^{-2}}{3}}A_j(q), \tag{29}$$

and that $\bar{A}_j(q)$ satisfies the equivariance relation

$$q^{\frac{S^z}{2}} \triangleright \bar{A}_j(q) = q^{\frac{S^z}{2}}\bar{A}_j(q)q^{-\frac{S^z}{2}}. \tag{30}$$

We then use these facts to recover the $q$AKLT ground state as follows,

$$
\begin{aligned}
M \triangleright |\text{AKLT}\rangle_{\alpha\beta} &= \big([M_1 \triangleright A_1(1)][M_2 \triangleright A_2(1)]\cdots[M_L \triangleright A_L(1)]\big)_{\alpha\beta} \\
&= \big(q^{-S^z-\frac{1}{2}}\bar{A}_1(q)\bar{A}_2(q)\cdots\bar{A}_L(q)q^{(2L+1)S^z+\frac{1}{2}}\big)_{\alpha\beta} \\
&= \big(q^{-S^z-1}\Xi A_1(q)\Xi A_2(q)\cdots\Xi A_L(q)q^{(2L+1)S^z+1}\big)_{\alpha\beta} \\
&= \Xi^L\big(q^{-S^z}A_1(q)A_2(q)\cdots A_L(q)q^{(2L+1)S^z}\big)_{\alpha\beta} \\
&= \Xi^L q^{-\alpha+(2L+1)\beta}|q\text{AKLT}\rangle_{\alpha\beta},
\end{aligned} \tag{32}
$$

---

[10]We use the symbol $\triangleright$ to stress that an action is on the physical index of the rank-3 tensor $A$.

where $\Xi = \sqrt{(1 + q^2 + q^{-2})/3}$ and we have assumed that the entries of the $2 \times 2$ matrix are referred to by the choice of boundary spins $\alpha, \beta \in \left\{\pm\frac{1}{2}\right\}$.

Now we derive the transformation between the Hamiltonians. Because the local AKLT Hamiltonian is a Hermitian projector, it can be written

$$H_{j,j+1}^{\text{AKLT}} = (H_{j,j+1}^{\text{AKLT}})^\dagger H_{j,j+1}^{\text{AKLT}}. \tag{33}$$

The $\mathfrak{su}(2)$ spin-2 basis states written in a tensor product decomposition of spin-1 basis states can be obtained by setting $q = 1$ in Equation (24),

$$|\psi_{\pm 2}\rangle = |\pm\pm\rangle, \qquad |\psi_{\pm 1}\rangle = \frac{1}{\sqrt{2}}\big(|\pm 0\rangle + |0\pm\rangle\big), \qquad |\psi_0\rangle = \frac{1}{\sqrt{6}}\big(2|00\rangle + |+-\rangle + |-+\rangle\big). \tag{34}$$

Conjugating $H_{j,j+1}^{\text{AKLT}} = \sum_{m=-2}^{2} |\psi_m\rangle\langle\psi_m|$ with the Witten's Conjugation operator, we obtain

$$M|\psi_{\pm 2}\rangle\langle\psi_{\pm 2}|M^{-1} = |\pm\pm\rangle\langle\pm\pm| = |\psi_{\pm 2}^{(q)}\rangle\langle\psi_{\pm 2}^{(q)}|,$$

$$M|\psi_{\pm 1}\rangle\langle\psi_{\pm 1}|M^{-1} = \frac{q^2 + q^{-2}}{2}|\psi_{\pm 1}^{(q^{-1})}\rangle\langle\psi_{\pm 1}^{(q)}|, \tag{35}$$

$$M|\psi_0\rangle\langle\psi_0|M^{-1} = \frac{1}{6}\sqrt{q^4 + q^{-4} + (q + q^{-1})^2}|\omega\rangle\langle\psi_0^{(q)}|,$$

where

$$|\omega\rangle = \frac{4}{q + q^{-1}}|00\rangle + q^2|+-\rangle + q^{-2}|-+\rangle. \tag{36}$$

Note that $|\omega\rangle$ is orthogonal to $|\psi_{\pm 1}^{(q^{-1})}\rangle$ and $|\psi_{\pm 2}^{(q)}\rangle$. Thus choosing $C_{j,j+1}$ in Equation (21) to be

$$\begin{aligned} C_{j,j+1} = &|\psi_2^{(q)}\rangle\langle\psi_2^{(q)}| + |\psi_{-2}^{(q)}\rangle\langle\psi_{-2}^{(q)}| \\ &+ \frac{4}{(q^2 + q^{-2})^2}\big(|\psi_1^{(q^{-1})}\rangle\langle\psi_1^{(q^{-1})}| + |\psi_{-1}^{(q^{-1})}\rangle\langle\psi_{-1}^{(q^{-1})}|\big) \\ &+ \frac{36(q + q^{-1})^4}{\big[16 + (q^4 + q^{-4})(q + q^{-1})\big]^2\big[q^4 + q^{-4} + (q + q^{-1})^2\big]}|\omega\rangle\langle\omega|, \end{aligned} \tag{37}$$

results in the $q$AKLT Hamiltonian (22). This operator is also Hermitian and positive definite as required.

It is also possible to perform the inverse transformation and obtain the AKLT Hamiltonian from the $q$AKLT Hamiltonian. The procedure is exactly the same, except we use $M^{-1}$ and make a different choice for the Hermitian degree of freedom in Equation (21). This operator is

$$\begin{aligned} D_{j,j+1} = &|\psi_2^{(q)}\rangle\langle\psi_2^{(q)}| + |\psi_{-2}^{(q)}\rangle\langle\psi_{-2}^{(q)}| \\ &+ \frac{(q^2 + q^{-2})^2}{2}\left[\frac{q^{-1}}{(q^{-2} + q^6)^2}|\omega_1\rangle\langle\omega_1| + \frac{q}{(q^2 + q^{-6})^2}|\omega_{-1}\rangle\langle\omega_{-1}|\right] \\ &+ \frac{6}{\big[q^4 + q^{-4} + (q + q^{-1})^2\big]^2\big[(q + q^{-1})^4 + 4(q^8 + q^{-8})\big]^2}|\omega_0\rangle\langle\omega_0|, \end{aligned} \tag{38}$$

where

$$\begin{aligned} |\omega_{\pm 1}\rangle &= q^{\mp 1}|\pm 0\rangle + q^{\pm 3}|0\pm\rangle, \\ |\omega_0\rangle &= (q + q^{-1})^2|00\rangle + 2\big(q^{-4}|+-\rangle + q^4|-+\rangle\big). \end{aligned} \tag{39}$$

It can be checked that indeed

$$H_{j,j+1}^{\text{AKLT}} = (M^{-1}H_{j,j+1}^{q\text{AKLT}}M)^\dagger D_{j,j+1}M^{-1}H_{j,j+1}^{q\text{AKLT}}M, \tag{40}$$

is satisfied with $D_{j,j+1}$ as defined in (38).

# 4 The $q$-deformed Kennedy-Tasaki duality

In this section we construct a $q$-deformed KT dual for the $q$AKLT Hamiltonian according to the commutative diagram

$$
\begin{array}{ccc}
H_{\mathrm{AKLT}} & \xrightarrow{\ U_{\mathrm{KT}}\ } & H_{\mathrm{SSB}} \\[4pt]
{\scriptstyle M^{-1}}\big\uparrow & & \big\downarrow{\scriptstyle M} \\[4pt]
H_{q\mathrm{AKLT}} & \dashrightarrow[\mathcal{N}_{q\mathrm{KT}}] & H_{q\mathrm{SSB}}
\end{array}
$$

In Section 4.1 we explain the process in detail and show that this construction fixes the locality issues discussed in Section 2.2. Section 4.2 is spent explaining how $H_{q\mathrm{SSB}}$ exhibits spontaneous $\mathbb{Z}_2 \times \mathbb{Z}_2^{(q)}$ symmetry-breaking.

## 4.1 The $q$-deformed Kennedy-Tasaki dual of the $q$AKLT model

Applying the KT transformation to the AKLT Hamiltonian results in another Hermitian nearest-neighbor projector Hamiltonian [16, 17], which we call $H_{\mathrm{SSB}}$. We want the same statement to hold for the $q$-deformed KT dual $H_{q\mathrm{SSB}}$. Hermiticity is guaranteed by the implementation of Witten's Conjugation, but the projector property is not. As outlined in Section 3, Witten's Conjugation is implemented via a basis change in constituent terms of the Hamiltonian plus the potential inclusion of additional degrees of freedom to fine-tune the Hamiltonian as desired.[11] We will use the degree of freedom in Witten's Conjugation to ensure the local terms in $H_{q\mathrm{SSB}}$ are indeed *Hermitian* projectors. This guarantees that our duality transformation preserves the spectrum and hermiticity of the local Hamiltonian. Note that we do not claim that the full excitation spectrum is the same as $H_{q\mathrm{AKLT}}$. Nonetheless we conjecture the existence of a spectral gap above the ground states. Proving the existence of such a gap is complicated by the lack of translation-invariance in $H_{q\mathrm{SSB}}$, meaning standard approaches such as Knabe's method [41] or even finite-size scaling cannot be directly applied. In Appendix D we outline a heuristic argument which supports the existence of a gap in $H_{q\mathrm{SSB}}$. Note that we again take $q > 0$ in constructing $H_{q\mathrm{SSB}}$.

In the undeformed case, the space of ground states of the Hamiltonian $H_{\mathrm{SSB}}$ is spanned by the four product states $\big\{|\Phi_{1,\pm}\rangle, |\Phi_{2,\pm}\rangle\big\}$ given in Equation (14). The local Hamiltonian is a projector onto the orthogonal complement of the space of two-site ground states $\mathrm{Span}_{\mathbb{C}}\{|\phi_{1,\pm}\rangle^{\otimes 2}, |\phi_{2,\pm}\rangle^{\otimes 2}\}$. The orthonormal set of states,

$$
\left\{ \sqrt{\frac{2}{3}}\Big[|00\rangle - \frac{1}{2}\big(|++\rangle + |--\rangle\big)\Big], \frac{1}{\sqrt{2}}\big(|\pm 0\rangle - |0\pm\rangle\big), |\pm \mp\rangle \right\}, \tag{41}
$$

is orthogonal to the states $\{|\phi_{1,\pm}\rangle^{\otimes 2}, |\phi_{2,\pm}\rangle^{\otimes 2}\}$. The space spanned by these states is then the orthogonal complement to the space of two-site ground states for $H_{\mathrm{SSB}}$. Thus the local Hamiltonian can be written

$$
\begin{aligned}
H_{j,j+1}^{\mathrm{SSB}} = {} & \frac{2}{3}\Big[|00\rangle - \frac{1}{2}\big(|++\rangle + |--\rangle\big)\Big]\Big[\langle 00| - \frac{1}{2}\big(\langle ++| + \langle --|\big)\Big] \\
& + \frac{1}{2}\big(|+0\rangle - |0+\rangle\big)\big(\langle +0| - \langle 0+|\big) + \frac{1}{2}\big(|-0\rangle - |0-\rangle\big)\big(\langle -0| - \langle 0-|\big) \\
& + |+-\rangle\langle +-| + |-+\rangle\langle -+|.
\end{aligned} \tag{42}
$$

It is possible to rewrite this expression in terms of spin operators. However, as this does not provide a benefit in our context we will refrain from doing so.

---

[11]These degrees of freedom correspond to the choice of positive-definite Hermitian operators $C_{j,j+1}$ in (21).

Because $H^{\text{SSB}}_{j,j+1}$ is a Hermitian projector, we have $H^{\text{SSB}}_{j,j+1} = (H^{\text{SSB}}_{j,j+1})^\dagger H^{\text{SSB}}_{j,j+1}$. So we can apply Witten's Conjugation to $H_{\text{SSB}} = \sum_{j=1}^{L-1} H^{\text{SSB}}_{j,j+1}$ using the Witten's Conjugation operator (26) as we did for the $q$AKLT Hamiltonian. We then turn the result into a projector Hamiltonian using the degree of freedom available in the deformation process.

First, we conjugate $H^{\text{SSB}}_{j,j+1}$ to obtain

$$
\begin{aligned}
L^{q\text{SSB}}_{j,j+1} &= M H^{\text{SSB}}_{j,j+1} M^{-1} \\
&= \frac{2}{3} \frac{1}{q+q^{-1}} \Big[ |00\rangle - \frac{q+q^{-1}}{4} \big( q^{-4j-2}|++\rangle + q^{4j+2}|--\rangle \big) \Big] \\
&\qquad \times \Big[ (q+q^{-1})\langle 00| - q^{4j+2}\langle ++| - q^{-4j-2}\langle --| \Big] \\
&\quad + \frac{1}{2} \big( |+0\rangle - q^{-2}|0+\rangle \big)\big( \langle +0| - q^2\langle 0+| \big) \\
&\quad + \frac{1}{2} \big( |-0\rangle - q^2|0-\rangle \big)\big( \langle -0| - q^{-2}\langle 0-| \big) \\
&\quad + |+-\rangle\langle +-| + |-+\rangle\langle -+| .
\end{aligned}
\tag{43}
$$

We get the ground states of our target Hamiltonian by applying $M$ to the states $\{|\phi_{1,\pm}\rangle^{\otimes L}, |\phi_{2,\pm}\rangle^{\otimes L}\}$ given in Equation (14). That is, for any Hamiltonian $H_{q\text{SSB}} = \sum_{j=1}^{L-1} (L^{q\text{SSB}}_{j,j+1})^\dagger C^{q\text{SSB}}_{j,j+1} L^{q\text{SSB}}_{j,j+1}$ the space of its ground states is guaranteed to be spanned by the product states

$$
|\Phi^{(q)}_{1,\pm}\rangle = \bigotimes_{j=1}^{L} |\phi^{(q)}_{1,\pm}\rangle_j, \quad \text{and} \quad |\Phi^{(q)}_{2,\pm}\rangle = \bigotimes_{j=1}^{L} |\phi^{(q)}_{2,\pm}\rangle_j,
\tag{44}
$$

where

$$
|\phi^{(q)}_{1,\pm}\rangle_j = \frac{1}{\sqrt{a_j(q^{\pm 1})}} \big( |0\rangle + q^{\mp 2j}\sqrt{q+q^{-1}}|\pm\rangle \big),
\tag{45a}
$$

$$
|\phi^{(q)}_{2,\pm}\rangle_j = \frac{1}{\sqrt{a_j(q^{\pm 1})}} \big( |0\rangle - q^{\mp 2j}\sqrt{q+q^{-1}}|\pm\rangle \big),
\tag{45b}
$$

and $a_j(q) = 1 + q^{-4j}(q+q^{-1})$. Note that each term in the product decomposition of the ground states depends on its position in the chain; this will be reflected in an absence of translation-invariance in the final Hamiltonian.[12]

We will use the Hermitian degree of freedom to make the local terms in $H_{q\text{SSB}}$ Hermitian projectors by writing down the orthogonal complement of the set of ground states and comparing the result with Equation (43). That is, we first require an orthonormal set of states which spans the orthogonal complement of the space $\text{Span}_{\mathbb{C}}\{|\phi^{(q)}_{1,\pm}\rangle_j |\phi^{(q)}_{1,\pm}\rangle_{j+1}, |\phi^{(q)}_{2,\pm}\rangle_j |\phi^{(q)}_{2,\pm}\rangle_{j+1}\}$ of two-site ground states. The states

$$
\Bigg\{ |\mu^{(q)}_{\pm 2}\rangle = |\pm\mp\rangle ,
$$

$$
|\mu^{(q)}_{\pm 1}\rangle = \frac{1}{\sqrt{q^2+q^{-2}}} \big( q^{\mp 1}|\pm 0\rangle - q^{\pm 1}|0\pm\rangle \big),
\tag{46}
$$

$$
|\mu^{(q)}_0(j)\rangle = \frac{1}{\sqrt{q^{8j+4}+q^{-8j-4}+(q+q^{-1})^2}} \big[ (q+q^{-1})|00\rangle - q^{4j+2}|++\rangle - q^{-4j-2}|--\rangle \big] \Bigg\},
$$

---

[12]The absence of translation invariance complicates the interpretation of correlation functions and order parameters. We discuss this briefly at the end of Section 4.2.

form such a set. Hence we desire that the local Hamiltonian takes the form

$$H_{j,j+1}^{q\text{SSB}} = |\mu_0^{(q)}(j)\rangle\langle\mu_0^{(q)}(j)| + |\mu_1^{(q)}\rangle\langle\mu_1^{(q)}| + |\mu_{-1}^{(q)}\rangle\langle\mu_{-1}^{(q)}| + |\mu_2^{(q)}\rangle\langle\mu_2^{(q)}| + |\mu_{-2}^{(q)}\rangle\langle\mu_{-2}^{(q)}|. \tag{47}$$

Note that there are site-dependent terms in the local Hamiltonian, so translation symmetry is broken. We discuss this feature more below and in Section 4.2. We could also write the Hamiltonian in terms of the spin operators, but again refrain from doing so as the resulting expression becomes unwieldy and does not offer additional insights.

In order to obtain $H_{j,j+1}^{q\text{SSB}} = (L_{j,j+1}^{q\text{SSB}})^\dagger C_{j,j+1}^{q\text{SSB}} L_{j,j+1}^{q\text{SSB}}$ as in Equation (47), we choose the Hermitian operator in (21) to be

$$\begin{aligned}
C_{j,j+1}^{q\text{SSB}} = {} & \frac{9}{4^3} \frac{(q+q^{-1})^2}{[q^{8j+4} + q^{-8j-4} + (q+q^{-1})^2][16 + (q+q^{-1})^2(q^{8j+4}+q^{-8j-4})]^2}|\xi\rangle\langle\xi| \\
& + \frac{4}{(q^2+q^{-2})^3}\Big[\big(q|+0\rangle - q^{-1}|0+\rangle\big)\big(q\langle+0| - q^{-1}\langle0+|\big) \\
& + \big(q^{-1}|-0\rangle - q|0-\rangle\big)\big(q^{-1}\langle-0| - q\langle0-|\big)\Big] \\
& + |+-\rangle\langle+-| + |-+\rangle\langle-+|, \tag{48}
\end{aligned}$$

where we have defined

$$|\xi\rangle = |00\rangle + \frac{4}{q+q^{-1}}\big(q^{4j+2}|++\rangle + q^{-4j-2}|--\rangle\big). \tag{49}$$

Note that $C_{j,j+1}^{q\text{SSB}}$ is positive-definite and Hermitian, as required.

Before discussing the symmetries of $H_{q\text{SSB}}$, let us briefly discuss its behavior in the thermodynamic limit. For the finite chain translation symmetry is broken by the appearance of the projector $|\mu_0^{(q)}(j)\rangle\langle\mu_0^{(q)}(j)|$ which is site-dependent. However, considering the far-left or right of the chain, these terms become fixed projectors. Taking $q > 1$ for example, we find

$$\lim_{j\to\pm\infty} |\mu_0^{(q)}(j)\rangle\langle\mu_0^{(q)}(j)| = |\pm\pm\rangle\langle\pm\pm|. \tag{50}$$

For $q \in (0,1)$, the projectors to the far left and far right satisfy Equation (50) with the replacement $\pm \mapsto \mp$ on the right hand side. Note that the cases $q < 1$ and $q > 1$ can be related by exchanging $q$ with $q^{-1}$, and in consistency with our characterization of $\mathbb{Z}_2 \times \mathbb{Z}_2^{(q)}$, applying $R^x$ or $R^y$ to (50). We discuss again the behavior of $H_{q\text{SSB}}$ in the thermodynamic limit at the level of the ground states in Section 4.2.

It can be readily checked that the local Hamiltonian (47) is not invariant under $U_q[\mathfrak{su}(2)]$ for general choices of $q$, nor under $\mathbb{Z}_2 \times \mathbb{Z}_2$, but it does possess $\mathbb{Z}_2 \times \mathbb{Z}_2^{(q)}$ invariance. We now show how exactly the preservation of $\mathbb{Z}_2 \times \mathbb{Z}_2^{(q)}$ invariance in $H_{q\text{SSB}}$ is guaranteed at each step of its construction. Let $O \xmapsto{R^\alpha} O'$ denote a $\pi$-rotation about the $\alpha$-axis of the operator $O$, which is implemented via conjugation by $R^\alpha = e^{i\pi S^\alpha}$. The KT operator $U_{\text{KT}}$ commutes with operations of $\mathbb{Z}_2 \times \mathbb{Z}_2$, and we observe that $\pi$-rotations about the $x$- or $y$-axis induce the transformation $M(q) \to M(q^{-1})$,

$$M_j(q) \xmapsto{R^x, R^y} M_j(q^{-1}), \qquad M_j(q) \xmapsto{R^z} M_j(q). \tag{51}$$

This implies $\mathcal{N}_{q\text{KT}}$ commutes with $\mathbb{Z}_2 \times \mathbb{Z}_2^{(q)}$ operations. We also need to address how the operator $C_{j,j+1}^{q\text{SSB}}$ transforms under $\mathbb{Z}_2 \times \mathbb{Z}_2^{(q)}$. Direct computation shows that

$$C_{j,j+1}^{q\text{SSB}} \xmapsto{R^x, R^y} C_{j,j+1}^{q^{-1}\text{SSB}}, \qquad C_{j,j+1}^{q\text{SSB}} \xmapsto{R^z} C_{j,j+1}^{q\text{SSB}}. \tag{52}$$

Hence $C_{j,j+1}^{q\text{SSB}}$ is invariant under $\mathbb{Z}_2 \times \mathbb{Z}_2^{(q)}$ operations, as desired.

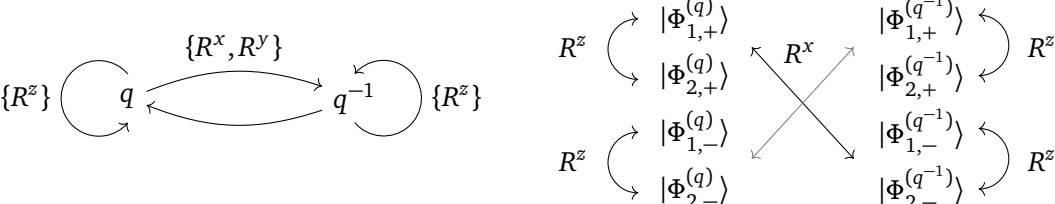

Figure 1: A diagrammatic presentation of spontaneous duality-symmetry breaking in $H_{q\text{SSB}}$. There is a model at $q$ and a model at $q^{-1}$, each invariant under $\mathbb{Z}_2 = \langle \mathbb{1}, R^z \rangle$ and possessing four ground states, with two $\mathbb{Z}_2$ orbits on each set of ground states. The set of $\pi$-rotations about the $x$- and $y$-axes are duality transformations between each model, and they take any state in the spin-up(down) orbit to the spin-down(up) orbit in the dual model.

## 4.2 Spontaneous duality-symmetry breaking

Because of the presence of the KT operator in $\mathcal{N}_{q\text{KT}} = M U_{\text{KT}} M^{-1}$ this transformation will map diluted antiferromagnetic order to ferromagnetic order. Since the ground states of the $q\text{AKLT}$ and $q\text{SSB}$ Hamiltonians are mapped to each other, up to normalization, by applying $\mathcal{N}_{q\text{KT}}$, we can conclude that each configuration of edge modes in the $q\text{AKLT}$ state is mapped to a distinct ferromagnetically ordered state. We now explain how the basis (45) can be used to demonstrate $\mathbb{Z}_2 \times \mathbb{Z}_2^{(q)}$ SSB in the finite chain, and discuss subtleties that arise in the infinite chain.

There are two $\mathbb{Z}_2 = \langle \mathbb{1}, R^z \rangle$ orbits in the set of ground states for $H_{q\text{SSB}}$, which we label with a subscript $+$ and $-$ inside the kets:

$$R^z|\Phi_{1,\pm}^{(q)}\rangle = |\Phi_{2,\pm}^{(q)}\rangle. \tag{53}$$

We call $\left\{ |\Phi_{1,+}^{(q)}\rangle, |\Phi_{2,+}^{(q)}\rangle \right\}$ the spin-up orbit and $\left\{ |\Phi_{1,-}^{(q)}\rangle, |\Phi_{2,-}^{(q)}\rangle \right\}$ the spin down orbit, since they correspond to ferromagnetically ordered spin-up and -down states respectively. Acting with $R^x$ or $R^y$ will take any state in the spin-up orbit of the model at $q$ to the spin-down orbit in the model at $q^{-1}$:

$$R^x|\Phi_{1,+}^{(q)}\rangle = -|\Phi_{1,-}^{(q^{-1})}\rangle, \qquad R^x|\Phi_{2,-}^{(q)}\rangle = -|\Phi_{2,+}^{(q^{-1})}\rangle, \tag{54}$$

and

$$R^y|\Phi_{1,+}^{(q)}\rangle = -|\Phi_{2,-}^{(q^{-1})}\rangle, \qquad R^y|\Phi_{1,-}^{(q)}\rangle = -|\Phi_{2,+}^{(q^{-1})}\rangle. \tag{55}$$

The set of ground states for $H_{q\text{SSB}}$ then form a $\mathbb{Z}_2 \times \mathbb{Z}_2^{(q)}$ orbit, and the union of the sets of ground states for $H_{q\text{SSB}}$ and $H_{q^{-1}\text{SSB}}$ form two separate $\mathbb{Z}_2 \times \mathbb{Z}_2$ orbits. See Figure 1 for a diagrammatic presentation of the various $\mathbb{Z}_2$ orbits.

We conclude from the above that the degeneracy in the ground states of $H_{q\text{SSB}}$ for a finite chain is a result of $\mathbb{Z}_2 \times \mathbb{Z}_2^{(q)}$ spontaneous symmetry breaking.

Although in the finite chain spontaneous symmetry breaking is manifest from the ground state degeneracy of $H_{q\text{SSB}}$ and the action of the duality-symmetry, the assignment of a local order parameter to measure this symmetry breaking in the thermodynamic limit is more complicated. The first complication is due to a lack of translation invariance which causes the expectation values of the two-point correlation functions to not depend solely on the distance $|i-j|$. In fact, the expectation value $S_i^z S_j^z$ in the ground states of $H_{q\text{SSB}}$ depends also on $|i+j|$.

The second complication is that the operator $\mathcal{N}_{q\mathrm{KT}}$ cannot be used to relate the string order parameter to a local order parameter as the KT transformation does in Equation (12). In particular, the operator

$$\mathcal{N}_{q\mathrm{KT}} S_i^z \prod_{k=i+1}^{j-1} e^{i\pi S_k^z} S_j^z \mathcal{N}_{q\mathrm{KT}}^\dagger \tag{56}$$

is non-local. This can be seen by using the final relation in (12) to remove the string of $e^{i\pi S_k^z}$ terms. After doing this we have $M U_{\mathrm{KT}} M^{-1} U_{\mathrm{KT}}$ and $U_{\mathrm{KT}} M^{-1} U_{\mathrm{KT}} M$ on the left and right of $S_i^z S_j^z$ respectively. Because $\mathbb{Z}_2 \times \mathbb{Z}_2$ acts non-trivially on $M$, it cannot give rise to a local interaction under the KT transformation, see Equation (51) and the discussion in Section 2.1. Explicitly, the non-local terms that appear are

$$M U_{\mathrm{KT}} M^{-1} U_{\mathrm{KT}} = U_{\mathrm{KT}} M^{-1} U_{\mathrm{KT}} M = \prod_{k=1}^{L} q^{-2k S_k^z} q^{2k \prod_{m=1}^{k-1} \exp(i\pi S_m^z) S_k^z} . \tag{57}$$

Hence removing the string term in (56) only leads to new non-local contributions from (57).

We note however that the local magnetization gives us some information about the order in the ground states of $H_{q\mathrm{SSB}}$ in the thermodynamic limit. Consider for example the expectation value of $S_j^z$,

$$m_\pm(j) = \langle \Phi_{1,\pm}^{(q)} | S_j^z | \Phi_{1,\pm}^{(q)} \rangle = \langle \Phi_{2,\pm}^{(q)} | S_j^z | \Phi_{2,\pm}^{(q)} \rangle = \pm \left( 1 - \frac{1}{1 + q^{\mp 4j}(q + q^{-1})} \right) . \tag{58}$$

By looking at the local magnetization on the far left and right of the chain in the thermodynamic limit, we observe that translation invariance is in some sense restored in these regions. First, for $q > 1$ we have

$$m_+(j) \to \begin{cases} 0, & \text{if } j \to \infty, \\ 1, & \text{if } j \to -\infty, \end{cases} \qquad m_-(j) \to \begin{cases} -1, & \text{if } j \to \infty, \\ 0, & \text{if } j \to -\infty. \end{cases} \tag{59}$$

The situation is similar for $q < 1$ with the roles of the limits $j \to \pm\infty$ being swapped. This agrees with what one would expect to observe from looking at which terms in the ground states (45) dominate in these limits. We observe from this that only the product states differing by the spin state (i.e. $\pm$ label) can be distinguished from each other amongst the set of ground states when looking at the far left or right of the chain. This is in contrast to the states differing by label 1 or 2, which give rise to the same order in the limits $j \to \pm\infty$.

## 5 Conclusions and discussion

We have revisited the connection between string order and hidden symmetry breaking in the $q$AKLT model with $SO_q(3)$ quantum group invariance. We showed that the Kennedy-Tasaki transformation generates non-local terms when applied to the $q$AKLT model for $q \neq 1$, questioning its interpretation as a duality transformation between an SPT and a SSB phase. The appearance of these non-local terms could be attributed to the deformation of the AKLT model's $\mathbb{Z}_2 \times \mathbb{Z}_2$ symmetry to a $\mathbb{Z}_2 \times \mathbb{Z}_2^{(q)}$ duality-symmetry which includes a change of parameter from $q$ to $q^{-1}$. Using Witten's Conjugation we then constructed a new duality transformation that resolves these locality issues. By studying the space of ground states for the dual Hamiltonian $H_{q\mathrm{SSB}}$, we showed that our dual model exhibits spontaneous $\mathbb{Z}_2 \times \mathbb{Z}_2^{(q)}$ symmetry breaking. We also explained in which sense the $\mathbb{Z}_2 \times \mathbb{Z}_2^{(q)}$ duality-symmetry is an automatic consequence of the quantum group symmetry, in close analogy to the undeformed case where $\mathbb{Z}_2 \times \mathbb{Z}_2$ is a subgroup of $SO(3)$ spin rotations.

While confirming an interesting picture of hidden symmetry breaking in the $q$AKLT model, our results have a slight technical limitation. The statements made in the previous paragraph implicitly assume the existence of a gap in the dual Hamiltonian $H_{q\text{SSB}}$. We provided strong evidence for the existence of this gap as long as the deformation parameter takes values in the finite interval $q \in (2^{-1/4}, 2^{1/4})$. However, we are currently not able to make predictions about the presence or absence of a gap for deformation parameters outside of this interval. While this clearly invites further investigations it does not limit the merit of the theory developed.

Our construction of the $q$-deformed duality was restricted to the case of spin-1 degrees of freedom. It is natural to ask whether this discussion can be extended to higher spins, see [30,42–45] for previous investigations of the corresponding $q$AKLT models. For our strategy to be applicable, we would require a suitable generalization of the Witten's Conjugation argument to higher spins. Alternatively, it may be possible to achieve the generalization more directly in one of the frameworks discussed below.

One remarkable feature of our duality transformation is that the SSB side exhibits an explicit breaking of translation invariance. This impeded the description of consistent order parameters capable of describing the hidden symmetry breaking which was therefore left for future study. Recent investigations have shown that the relation between symmetries, dualities and translation symmetries is surprisingly subtle even for relatively simple theories such as the transverse field Ising model [46–48]. It would be interesting to investigate whether the non-trivial topological properties of the $q$AKLT model and its hidden symmetry breaking can also be understood from a similar consideration of topological defect lines.

We used the local magnetization to analyze the order of the ground states of $H_{q\text{SSB}}$ in the thermodynamic limit. Our discussion revealed different behavior on the far left and far right of the spin chain, which was dictated by whether $q$ was greater or smaller than one. The behavior on the left and right is then exchanged upon sending $q \to q^{-1}$. This is natural when considering the action of inversion transformations on the $q$AKLT Hamiltonian, which also needs to be paired with the map $q \to q^{-1}$ in order to be a symmetry [30, 31]. While in this work we studied these features in order to understand the nature of hidden symmetry breaking in our Hamiltonians, it would be interesting to explore the effect of translation symmetry breaking on the infinite chain in more detail.

It remains to be understood whether and how our proposed strategy can apply to a broader phase diagram away from the $q$AKLT point. The Hamiltonian $H_{q\text{SSB}}$ should be viewed as a representative of the SSB phase dual to the SPT phase in which $H_{q\text{AKLT}}$ lies. This viewpoint is justified by the fact that the properties of $H_{q\text{SSB}}$ are exactly those that one expects in the dual SSB phase obtained from a Kennedy-Tasaki duality: The edge states in $H_{q\text{AKLT}}$ are mapped to product states and SPT order is mapped to SSB order. This is in fact guaranteed from the presence of the KT transformation in $\mathcal{N}_{q\text{KT}}$ and the $\mathbb{Z}_2 \times \mathbb{Z}_2^{(q)}$-invariance without $\mathbb{Z}_2 \times \mathbb{Z}_2$-invariance in $H_{q\text{SSB}}$. Phrasing the quantum group symmetry in terms of topological defect lines may be a good starting point to establish statements that apply to a broader phase diagram. Alternatively, one could try to apply one of the existing general schemes for exploring dualities. A systematic approach to dualities has been proposed by Cobanera et al. in [35]. In their framework, two Hamiltonians are said to be dual to each other if, roughly speaking, there is an isomorphism between their local constituents, commonly referred to as bonds. As individual bonds can be combined in different ways, this enables the discussion of whole families of Hamiltonians that are related by duality. The formalism requires that a chosen representative set of local operators must form a von Neumann algebra, the so-called bond algebra. Since one can formulate unitary implementations of von Neumann algebra isomorphisms, the duality transformation between Hamiltonians can be unitarily implemented under these circumstances [35]. This approach has been widely adopted [22, 37–39].

Recent years have also seen systematic approaches which utilise bond algebra isomorphisms while taking into account the relevant symmetries at play. We are aware of two such approaches that have been applied to concrete spin systems. Given the MPS construction of the $q$AKLT ground state, the use of matrix product operator intertwiners between bond algebras is particularly relevant [37, 38]. This requires knowledge of the relevant symmetry category for a given family of models, which in the case of the usual Haldane phase (with $q = 1$) is $\text{Vec}_{\mathbb{Z}_2 \times \mathbb{Z}_2}$. We do not have this data for the $q$AKLT model as it seems to fall outside of the usual framework of categorical symmetries [49]. Another approach is that of topological holography [22]. Topological holography also allows one to construct and relate local and non-local order parameters for the SPT and SSB phases, resolving the issues brought up at the end of Section 4.2. One would need to extend the topological holography formalism to apply to systems with quantum group symmetry in order to utilise this method as it is currently only applicable to systems with finite Abelian group symmetry (see however [50]).

An obstruction to constructing our transformation via the bond algebra approach is that the spectrum is not necessarily preserved in our mapping, so the bond algebras for $H_{q\text{AKLT}}$ and $H_{q\text{SSB}}$ cannot be isomorphic. In applying the bond algebra formalism one must choose a set of minimal $\mathbb{Z}_2 \times \mathbb{Z}_2^{(q)}$-symmetric couplings to be used as a bond algebra, and what that choice should be should be is not clear to us at this point in time. We remark that the authors of Ref. [37] consider Hamiltonians with $\text{Rep}(U_q[\mathfrak{su}(2)])$ matrix product operator symmetry where the choice of bonds is naturally given in terms of Clebsch-Gordan coefficients for $U_q[\mathfrak{su}(2)]$. However, the global symmetry in our model of interest is $U_q[\mathfrak{su}(2)]$ itself and not its category of modules. We also note that the bond algebra approach is designed to realize dualities directly in the thermodynamic limit while it is known that the notion of quantum group symmetry leads to substantial mathematical subtleties in the conventional $C^*$-algebraic approach to infinite quantum spin systems [51] (see however [52]).

We would like to note that we worked with open boundary conditions for the entirety of our discussion. Recently, the original Kennedy-Tasaki transformation has been implemented on a closed periodic chain [23]. In such a situation the Kennedy-Tasaki operator becomes a non-invertible defect implementing "twisted gauging" of the $\mathbb{Z}_2 \times \mathbb{Z}_2$ symmetry [23]. It would be interesting to establish a similar correspondence for the $q$-deformed duality.

## Acknowledgments

TF would like to thank Yuhan Gai for discussions on closely related topics. TQ would like to thank Robert Pryor for useful discussions and collaboration on a closely related project.

## A    The quantum group $U_q[\mathfrak{su}(2)]$ and its representations

Our definition for $U_q[\mathfrak{su}(2)]$ follows Chapter 3 of [53].[13] The information given here is also summarized in Appendices A and B of [30].

We define the *q-numbers* to be

$$[x]_q = \frac{q^x - q^{-x}}{q - q^{-1}} = [x]_{q^{-1}}. \tag{A.1}$$

Note that $\lim_{q \to 1} [x]_q = x$. We allow $x$ to be an operator or a complex number.

The quantum group $U_q[\mathfrak{su}(2)]$ is a Hopf algebra generated by $\mathbb{S}^z$, $\mathbb{S}^\pm$, with commutation relations

$$[\mathbb{S}^z, \mathbb{S}^\pm] = \pm \mathbb{S}^\pm, \quad \text{and} \quad [\mathbb{S}^+, \mathbb{S}^-] = [2\mathbb{S}^z]_q. \tag{A.2}$$

---

[13]Our $U_q[\mathfrak{su}(2)]$ generators are related to the ones in [53] by $K^\pm = q^{\pm 2\mathbb{S}^z}$, $E = \mathbb{S}^+$, $F = \mathbb{S}^-$.

The Hopf algebra structure is given by the counit $\varepsilon : U_q[\mathfrak{su}(2)] \to \mathbb{C}$, antipode $S : U_q[\mathfrak{su}(2)] \to U_q[\mathfrak{su}(2)]$, and coproduct $\Delta : U_q[\mathfrak{su}(2)] \to U_q[\mathfrak{su}(2)] \otimes U_q[\mathfrak{su}(2)]$:

$$\varepsilon(\mathbb{S}^z) = \varepsilon(\mathbb{S}^\pm) = 0, \tag{A.3a}$$

$$S(\mathbb{S}^z) = -\mathbb{S}^z, \quad S(\mathbb{S}^\pm) = -q^{\pm 1}\mathbb{S}^\pm, \tag{A.3b}$$

$$\Delta(\mathbb{S}^z) = \mathbb{S}^z \otimes \mathbb{1} + \mathbb{1} \otimes \mathbb{S}^z, \tag{A.3c}$$

$$\Delta(\mathbb{S}^\pm) = \mathbb{S}^\pm \otimes q^{\mathbb{S}^z} + q^{-\mathbb{S}^z} \otimes \mathbb{S}^\pm. \tag{A.3d}$$

The structure of irreducible representations of $U_q[\mathfrak{su}(2)]$ for generic $q$ (which for us means $q > 0$ and finite) has many similarities to that of $\mathfrak{su}(2)$. An irreducible spin-$S$ representation of $U_q[\mathfrak{su}(2)]$, denoted $\mathcal{V}_S$, is $(2S + 1)$-dimensional and has an orthonormal basis set $\{|m\rangle\}_{m=-S}^S$, on which $U_q[\mathfrak{su}(2)]$ is represented via the relations

$$\mathbb{S}^z|m\rangle = m|m\rangle,$$

$$\mathbb{S}^+|m\rangle = \begin{cases} \sqrt{[S-m]_q[S+m+1]_q}\,|m+1\rangle, & \text{if} \quad m < S, \\ 0, & \text{if} \quad m = S, \end{cases}$$

$$\mathbb{S}^-|m\rangle = \begin{cases} \sqrt{[S+m]_q[S-m+1]_q}\,|m-1\rangle, & \text{if} \quad m > -S, \\ 0, & \text{if} \quad m = -S. \end{cases} \tag{A.4}$$

Taking $q \to 1$ we completely recover $\mathfrak{su}(2)$ or rather its universal enveloping algebra $U[\mathfrak{su}(2)]$. Note that we can use the standard basis of $\mathbb{C}^{2S+1}$ as the $\mathbb{S}^z$ eigenbasis, where $|m\rangle$ is the vector with its $m^{\text{th}}$ entry being 1 and all others zero. We can then identify $\mathbb{S}^z = S^z$ in this basis.

In certain representations the $U_q[\mathfrak{su}(2)]$ generators are proportional to the ordinary $\mathfrak{su}(2)$ generators. For $S = 1/2$, the relations (A.2) are satisfied by

$$\mathbb{S}^z = S^z, \quad \text{and} \quad \mathbb{S}^\pm = S^\pm, \tag{A.5}$$

and for $S = 1$ we can take

$$\mathbb{S}^z = S^z, \quad \text{and} \quad \mathbb{S}^\pm = \sqrt{\frac{q + q^{-1}}{2}} S^\pm. \tag{A.6}$$

For higher spin the $U_q[\mathfrak{su}(2)]$ generators cease to be proportional to the $\mathfrak{su}(2)$ generators.

As spin systems are represented in a complex Hilbert space, we require dual representations, determined by the Hopf-$*$ algebra structure:

$$(\mathbb{S}^z)^* = \mathbb{S}^z, \qquad (\mathbb{S}^\pm)^* = \mathbb{S}^\mp. \tag{A.7}$$

Note that the Hopf-$*$ algebra structure of $U_q[\mathfrak{su}(2)]$ is very similar to the structure of Hermitian adjoints of representations of $\mathfrak{su}(2)$.

## B  Quantum group symmetry implies duality symmetry

Here we expand upon the remark made in Section 2.2 that $\mathbb{Z}_2 \times \mathbb{Z}_2^{(q)}$ duality-symmetry is inherent in $U_q[\mathfrak{su}(2)]$-invariant Hamiltonians. We focus our discussion on nearest-neighbor Hamiltonians for concreteness.

First consider the action of $\mathbb{Z}_2 \times \mathbb{Z}_2$ generated by $\pi$-rotations about the principal axes on the $U_q[\mathfrak{su}(2)]$ generators in integer spin representations. We write $O \xrightarrow{R^\alpha} O'$ to denote a $\pi$-rotation about the $\alpha$-axis. This is implemented via conjugation by $e^{i\pi S^\alpha}$. On the ordinary $SU(2)$ spin generators $\vec{S} = (S^x, S^y, S^z)$, we have

$$\vec{S} \xrightarrow{R^x} (S^x, -S^y, -S^z), \qquad \vec{S} \xrightarrow{R^y} (-S^x, S^y, -S^z), \qquad \vec{S} \xrightarrow{R^z} (-S^x, -S^y, S^z). \tag{B.1}$$

Note that this implies

$$S^{\pm} \xrightarrow{R^x} S^{\mp}, \qquad S^{\pm} \xrightarrow{R^y} -S^{\mp}, \qquad S^{\pm} \xrightarrow{R^z} -S^{\pm}, \tag{B.2}$$

where $S^{\pm} = S^x \pm iS^y$ are the $\mathfrak{su}(2)$ ladder operators.

It turns out that the same transformation properties hold for the generators of the quantum group $U_q[\mathfrak{su}(2)]$ even though the derivation is more subtle as it cannot be reduced to geometric intuition. To be specific, the transformations under $\pi$-rotations are as follows,

$$\mathbb{S}^z \xrightarrow{R^x, R^y} -\mathbb{S}^z, \qquad \mathbb{S}^z \xrightarrow{R^z} \mathbb{S}^z, \tag{B.3a}$$

$$\mathbb{S}^{\pm} \xrightarrow{R^x} \mathbb{S}^{\mp}, \qquad \mathbb{S}^{\pm} \xrightarrow{R^y} -\mathbb{S}^{\mp}, \qquad \mathbb{S}^{\pm} \xrightarrow{R^z} -\mathbb{S}^{\pm}. \tag{B.3b}$$

To prove these relations consider an integer spin-$S$ representation and choose the standard basis for $\mathbb{C}^{2S+1}$ as described after Equation (A.4). In this basis we have $\mathbb{S}^z \equiv S^z$, so that the transformations (B.3a) follow immediately. The $\pi$-rotation about the $z$-axis then acts like

$$e^{i\pi S^z}|m\rangle = (-1)^m|m\rangle, \tag{B.4}$$

on basis vectors. Combining this with the action of $\mathbb{S}^{\pm}$ given in Equation (A.4), implies the last relation in (B.3b). The action of $\pi$-rotations about the $x$- and $y$-axes on basis vectors can be found to be

$$e^{i\pi S^x}|m\rangle = (-1)^S|-m\rangle, \qquad e^{i\pi S^y}|m\rangle = (-1)^{S+m}|-m\rangle, \tag{B.5}$$

by expanding the spin-$S$ basis vectors in terms of a tensor product of $2S$ spin-$\frac{1}{2}$ basis vectors. See Problem 2.1.g in Ref. [54] for a detailed explanation of how this is done. Combining Equation (B.5) with the action of $\mathbb{S}^{\pm}$ given in Equation (A.4) and using a symmetry property of the normalization factors, we can finally confirm the first two relations in Equation (B.3b).

Alternatively, we can use the actions in Equation (B.3) to realise $\mathbb{Z}_2 \times \mathbb{Z}_2^{(q)}$ purely at the algebraic level as an automorphism of $U_q[\mathfrak{su}(2)]$. To see this, note that these actions preserve the $U_q[\mathfrak{su}(2)]$ commutation relations (A.2) and the Hopf algebra structure (A.3) when paired with sending $q$ to $q^{-1}$ for the transformations $R^x$ and $R^y$. This is completely analogous to the $\mathbb{Z}_2 \times \mathbb{Z}_2$ automorphism of $\mathfrak{su}(2)$ via the actions (B.1).

Now consider how $\pi$-rotations act on $U_q[\mathfrak{su}(2)]$-invariant Hamiltonians. In analogy to the standard construction of $SU(2)$ Hamiltonians, we construct $U_q[\mathfrak{su}(2)]$-invariant Hamiltonians in polynomials of the Casimir acting locally. The $U_q[\mathfrak{su}(2)]$ quadratic Casimir is

$$C_q = \mathbb{S}^-\mathbb{S}^+ + \left(\left[\mathbb{S}^z + \frac{1}{2}\right]_q\right)^2 - \left(\left[\frac{1}{2}\right]_q\right)^2. \tag{B.6}$$

A straightforward calculation shows that each $\pi$-rotation leaves the Casimir invariant,

$$C_q \xrightarrow{R^\alpha} C_q, \quad \text{for} \quad \alpha = x, y, z. \tag{B.7}$$

On the other hand, the quadratic Casimir is invariant under sending $q$ to $q^{-1}$ as the $q$-numbers are invariant under this replacement. Hence $U_q[\mathfrak{su}(2)]$-invariant Hamiltonians with local terms that act on a *single* spin are $\mathbb{Z}_2 \times \mathbb{Z}_2$-invariant *as well as* $\mathbb{Z}_2 \times \mathbb{Z}_2^{(q)}$-invariant.

This ceases to be the case for $U_q[\mathfrak{su}(2)]$-invariant nearest-neighbor Hamiltonians. They will instead only be invariant under the $\mathbb{Z}_2 \times \mathbb{Z}_2^{(q)}$ duality-symmetry featuring in the main text. We see this by looking at the Casimir on a tensor product of irreducible $U_q[\mathfrak{su}(2)]$.

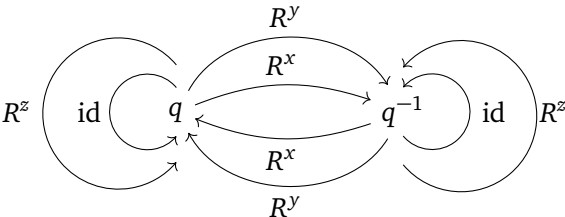

Figure 2: The quiver corresponding to the groupoid associated with $\mathbb{Z}_2 \times \mathbb{Z}_2^{(q)}$. Instead of forming an automorphism on the system, the rotations $R^x$ and $R^y$ perform the transformation $q \mapsto q^{-1}$, leaving all else unchanged. For the $q$AKLT model, the topological features are essentially the same for the $q$ and $q^{-1}$ systems, including at the point where $q = q^{-1} = 1$.

This is obtained by applying the coproduct to the single-site Casimir,

$$
\begin{aligned}
\Delta(C_q) &= \left(\mathbb{S}^- \otimes q^{\mathbb{S}^z} + q^{-\mathbb{S}^z} \otimes \mathbb{S}^-\right)\left(\mathbb{S}^+ \otimes q^{\mathbb{S}^z} + q^{-\mathbb{S}^z} \otimes \mathbb{S}^+\right) \\
&\quad + \left(\left[\mathbb{S}^z \otimes \mathbb{1} + \mathbb{1} \otimes \mathbb{S}^z + \frac{1}{2}\right]_q\right)^2 - \left(\left[\frac{1}{2}\right]_q\right) \\
&= \mathbb{S}^-\mathbb{S}^+ \otimes q^{2\mathbb{S}^z} + q^{-2\mathbb{S}^z} \otimes \mathbb{S}^-\mathbb{S}^+ + q^{-\mathbb{S}^z}\mathbb{S}^- \otimes q^{\mathbb{S}^z}\mathbb{S}^+ + q^{-\mathbb{S}^z}\mathbb{S}^+ \otimes q^{\mathbb{S}^z}\mathbb{S}^- \\
&\quad + \left(\left[\mathbb{S}^z \otimes \mathbb{1} + \mathbb{1} \otimes \mathbb{S}^z + \frac{1}{2}\right]_q\right)^2 - \left(\left[\frac{1}{2}\right]_q\right).
\end{aligned}
\tag{B.8}
$$

$U_q[\mathfrak{su}(2)]$-invariant nearest-neighbor Hamiltonians are polynomials in $\Delta(C_q)$. Acting with the $\pi$-rotations about the principal axes, we find

$$
\Delta(C_q) \xrightarrow{R^x, R^y} \Delta(C_{q^{-1}}), \qquad \Delta(C_q) \xrightarrow{R^z} \Delta(C_q).
\tag{B.9}
$$

For general values of $q$, the two-site $U_q[\mathfrak{su}(2)]$ Casimir, and thus any $U_q[\mathfrak{su}(2)]$-invariant nearest-neighbor Hamiltonian is then $\mathbb{Z}_2 \times \mathbb{Z}_2^{(q)}$-invariant but not $\mathbb{Z}_2 \times \mathbb{Z}_2$-invariant.

Treating the operation of sending $q$ to $q^{-1}$ as a symmetry transformation is natural in the context of $U_q[\mathfrak{su}(2)]$. We see this by noting that $U_q[\mathfrak{su}(2)]$ and $U_{q^{-1}}[\mathfrak{su}(2)]$ are isomorphic as algebras from (A.2). The reason why there is a distinction made between $q$ and $q^{-1}$ on tensor products of irreducible representations is because their coproducts, namely those for $\mathbb{S}^\pm$ (A.3d), differ. The coproduct for $U_{q^{-1}}[\mathfrak{su}(2)]$ is the opposite coproduct of $U_q[\mathfrak{su}(2)]$: it is the $U_q[\mathfrak{su}(2)]$ coproduct composed with the braiding operation on irreducible $U_q[\mathfrak{su}(2)]$ representations $\tau : V \otimes W \to W \otimes V$. However, the quantum group $U_q[\mathfrak{su}(2)]$ has a universal $R$-matrix, which implements the braiding as isomorphims which can be written in terms of the $U_q[\mathfrak{su}(2)]$ generators. In summary, $U_q[\mathfrak{su}(2)]$ and $U_{q^{-1}}[\mathfrak{su}(2)]$ are isomorphic as algebras, and this isomorphism is non-trivial for tensor products of irreducible representations. Thus the step sending $q$ to $q^{-1}$ in $\mathbb{Z}_2 \times \mathbb{Z}_2^{(q)}$ transformations should be considered as a symmetry transformation in the context of $U_q[\mathfrak{su}(2)]$.

We remark that the action of $\mathbb{Z}_2 \times \mathbb{Z}_2$ on the two-site Casimir (B.8) can be formulated as a groupoid, in that it is a set of symmetry transformations that acts between two objects. These objects are labelled by $q$ and $q^{-1}$. In addition to the identity morphisms there are morphisms $R^x, R^y \in \text{Hom}(q^{\pm 1}, q^{\mp 1})$ and $R^z \in \text{Hom}(q^{\pm 1}, q^{\pm 1})$. These morphisms compose according to the group composition rules for $\mathbb{Z}_2 \times \mathbb{Z}_2$. See Figure 2 for a presentation of this groupoid as a quiver.

It would be interesting to investigate the duality groups associated with more general quantum groups $U_q[\mathfrak{g}]$, where $\mathfrak{g}$ is a simple Lie algebra, thereby generalizing the results of [20, 21].

## C  The $q$AKLT ground state

In what follows raised indices take spin-1 configuration values, lowered take spin-1/2 configuration values, and all repeated indices are summed over. Our method of constructing the $q$AKLT MPS tensor follows Appendix C of Ref. [30].

Because the $q$AKLT Hamiltonian, given in Equations (16) and (22), is a sum over local projectors, we can construct the ground state as a *matrix product state* (MPS).[14] The MPS formalism is a convenient way of constructing a state that is in the kernel of each local projector. This is achieved by constructing each physical spin-1 from two auxiliary spin-1/2 irreducible $U_q[\mathfrak{su}(2)]$ representations. Each pair of auxiliary spins between physical sites is then projected onto the $U_q[\mathfrak{su}(2)]$ singlet,

$$|I^{(q)}\rangle = \frac{1}{\sqrt{q+q^{-1}}}\big(q^{\frac{1}{2}}|\uparrow\downarrow\rangle - q^{-\frac{1}{2}}|\downarrow\uparrow\rangle\big) = I_{\mu\nu}|\mu\nu\rangle. \tag{C.1}$$

The resulting state is projected onto the physical spin-1 lattice using the projector

$$P(q) = |+\rangle\langle\uparrow\uparrow| + \frac{1}{\sqrt{q+q^{-1}}}|0\rangle\big(q^{-\frac{1}{2}}\langle\uparrow\downarrow| + q^{\frac{1}{2}}\langle\downarrow\uparrow|\big) + |-\rangle\langle\downarrow\downarrow| = P^{\sigma}_{ab}|\sigma\rangle\langle ab|. \tag{C.2}$$

We construct the components of the MPS tensor using one physical spin-1 site, which is split into two auxiliary spin-1/2 sites, and the left auxiliary spin-1/2 site of the right neighboring spin-1. In other words, the MPS tensor is constructed by looking at $\mathcal{V}_{1/2} \otimes \mathcal{V}_{1/2} \otimes \mathcal{V}_{1/2}$, where the two left spin-1/2 irreducible representations belong to the same physical spin-1. The components of the (yet to be normalized) MPS tensor are defined by

$$\tilde{A}^{\sigma}_{\alpha\beta}|\sigma\rangle|\beta\rangle = \big(P(q)\otimes 1\big)|\alpha\rangle \otimes |I^{(q)}\rangle = P^{\sigma}_{\alpha\mu}I_{\mu\beta}|\sigma\rangle|\beta\rangle, \tag{C.3}$$

where $\alpha \in \{\uparrow, \downarrow\}$. So the components are written by identifying $\tilde{A}^{\sigma}_{\alpha\beta} = P^{\sigma}_{\alpha\mu}I_{\mu\beta}$. We often refer to the raised index as the physical index of the MPS tensor and the lowered indices as the virtual indices. Letting $\tilde{A}_j$ be the rank-3 tensor with components $\tilde{A}^{\sigma_j}_{\alpha\beta}$ with $|\sigma_j\rangle$ denoting the spin at site $j$, we write $\tilde{A}_j$ as a vector-valued matrix in the standard spin-1/2 basis,

$$\tilde{A}_j = \frac{1}{\sqrt{q+q^{-1}}}\begin{pmatrix} -\frac{q^{-1}}{\sqrt{q+q^{-1}}}|0\rangle_j & q^{\frac{1}{2}}|+\rangle_j \\ -q^{-\frac{1}{2}}|-\rangle_j & \frac{q}{\sqrt{q+q^{-1}}}|0\rangle_j \end{pmatrix}. \tag{C.4}$$

$\tilde{A}_j$ is right canonical but not normalized. The MPS tensor,

$$A_j = \frac{q+q^{-1}}{\sqrt{1+q^2+q^{-2}}}\tilde{A}_j, \tag{C.5}$$

is normalized and thus

$$|q\text{AKLT}\rangle_{\alpha\beta} = (A_1 A_2 \cdots A_L)_{\alpha\beta}, \tag{C.6}$$

are the normalized ground states of the AKLT Hamiltonian with open boundary conditions, where $\alpha$ and $\beta$ denote the spin-$\frac{1}{2}$ edge degrees of freedom.

---

[14]See Ref. [33] for a comprehensive review of the properties of matrix product states.

# D   Spectral gap of $H_{q\mathrm{SSB}}$

In order to support the validity of our hidden symmetry breaking picture, we outline a heuristic argument and show evidence that strongly suggests the existence of a gap in the spectrum of $H_{q\mathrm{SSB}}$ for (at least) the full interval $q \in (2^{-1/4}, 2^{1/4})$. To do this, we adopt Knabe's method [41], which allows one to compute a lower bound for the gap of a projector Hamiltonian in the thermodynamic limit by looking at finite subsystems. First we will briefly review Knabe's method, and then we introduce our heuristic argument and evidence.

Consider a translation-invariant Hamiltonian with periodic boundary condtions

$$H = \sum_{j=1}^{L} P_{j,j+1}, \tag{D.1}$$

where $P_{j,j+1}$ is a projector. If there exists $\epsilon > 0$ so that $H$ satisfies $H^2 \geq \epsilon H$, then $H$ has a spectral gap of at least $\epsilon$ in the thermodynamic limit. Consider now a finite length-$n$ subsystem

$$h_{n,i} = \sum_{j=i}^{i+n-1} P_{j,j+1}. \tag{D.2}$$

This system possesses a spectral gap, so $h_{n,i}^2 \geq \epsilon_n h_{n,i}$ for some $\epsilon_n > 0$. Knabe established the inequality

$$H^2 \geq \frac{n}{n-1}\left(\epsilon_n - \frac{1}{n}\right)H. \tag{D.3}$$

If there is some $n$ where $\epsilon_n > 1/n$, then the coefficient on the right hand side is positive and provides a lower bound on the gap $\epsilon$ of $H$. Notably, this bound is independent of the total system size so the gap of $H$ persists in the thermodynamic limit.

The Hamiltonian $H_{q\mathrm{SSB}}$ we are interested in is not translation-invariant, so the gap in the length-$n$ subsystem may also depend on which site the subsystem begins. Thus Knabe's method is, a priori, not applicable. Instead we will adopt the perspective that we can get meaningful information about the gap if we apply the method to the "tail ends" of the Hamiltonian at $j \to \pm\infty$ where the interactions become asymptotically site-independent (see Section 4.1). This seems reasonable as every finite system exhibits a gap and the possible absence of a gap is thus exclusively a large system size phenomenon. We then apply Knabe's method to these limiting Hamiltonians to obtain some evidence whether $H_{q\mathrm{SSB}}$ is gapped in the thermodynamic limit. To phrase it differently, we are rigorously establishing gaps for the tail end Hamiltonians for a finite (and coinciding) interval of the parameter $q$ and we claim that this implies the existence of a gap for the full Hamiltonian on the same interval.

The starting point for our investigation are the two local Hamiltonians

$$H_{i,i+1}^{\pm\infty} = |\pm\pm\rangle\langle\pm\pm| + |\mu_1^{(q)}\rangle\langle\mu_1^{(q)}| + |\mu_{-1}^{(q)}\rangle\langle\mu_{-1}^{(q)}| + |\mu_2^{(q)}\rangle\langle\mu_2^{(q)}| + |\mu_{-2}^{(q)}\rangle\langle\mu_{-2}^{(q)}|, \tag{D.4}$$

which arise by taking the limits $j \to \pm\infty$ of the Hamiltonian $H_{j,j+1}^{q\mathrm{SSB}}$ from Equation (47) (see Equation (50)). For simplicity and merely to make the case, we will consider the inequality (D.3) obtained by looking at a subsystem of *three* sites ($n = 3$),

$$h_{3,i}^{\pm} = H_{i,i+1}^{\pm\infty} + H_{i+1,i+2}^{\pm\infty}. \tag{D.5}$$

The Hamiltonian $h_{3,i}^{\pm}$ can be diagonalized, and the spectrum of $h_{3,i}^{+}$ is equal to that of $h_{3,i}^{-}$. The first excited state energy for both systems is

$$\epsilon_3 = \begin{cases} \frac{q^4}{1+q^4}, & \text{if } q > 1, \\ \frac{1}{1+q^4}, & \text{if } q < 1. \end{cases} \tag{D.6}$$

Knabe's Theorem predicts a gap in the thermodynamic limit for $\epsilon_3 > 1/3$. This proves that the auxiliary spin chains formed from employing the interactions $H_{i,i+1}^{\pm\infty}$ all along the lattice are gapped for $q \in (2^{-1/4}, 2^{1/4})$. Via our heuristic argument that these interactions should dominate the behavior of $H_{q\mathrm{SSB}}$, we use this as evidence to conjecture a gap in $H_{q\mathrm{SSB}}$ for the same values of $q$.

We finally note that the inequality we used does not predict the absence of a gap outside of the interval $q \in (2^{-1/4}, 2^{1/4})$ but is merely inconclusive. It is thus certainly possible that the Hamiltonian $H_{q\mathrm{SSB}}$ is gapped for a much wider range of $q$, potentially even all non-negative real values, and this is what we would expect. However, the investigation of the inequality (D.3) for larger (accessible) values of $n$, perhaps surprisingly, did not allow us to draw stronger conclusions.

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
