# Peer review of "Duality and hidden symmetry breaking in the q-deformed Affleck-Kennedy-Lieb-Tasaki model"

_SciPost Physics Core, doi:SciPost Phys. Core 7, 078 (2024)_

## Round 1 · Author Response

We thank both referees for carefully reading our manuscript and putting forward insightful comments, questions and suggestions.
Before responding to the referee comments individually we respond to the queries from the second reviewer regarding general applicability that were also raised by the editor: The $q$AKLT model is indeed finely-tuned so that one may construct the ground state as a matrix product state, however the physical properties, in particular string order, are part of a broader phase diagram [T. Quella Phys. Rev. B 103, 054404, 2021]. We adopt the perspective that $H_{q\text{AKLT}}$ and $H_{q\text{SSB}}$ serve as representatives of $q$-deformed SPT and SSB phases respectively. Our transformation cannot be used as a duality transformation for the entire $q$-deformed Haldane phase, however $H_{q\text{SSB}}$ indeed portrays the patterns of spontaneous symmetry breaking that we expect to hold in the dual phase. We have clarified this view in the introduction (page 3) and in the final paragraph on page 8, as well as added a further comment to this effect on pages 17-18.
Our responses to the individual comments from each referees are written below, and the changes listed appear in red in the updated manuscript.
--Responses to referee report 1--
-Possibility of a symmetry-protected trivial phase-
Comment: It is briefly mentioned in the Conclusion that the exchange of $q$ and $q^{-1}$ corresponds to the action of inversion transformation on the $q$AKLT Hamiltonian. An immediate consequence is that this Hamiltonian is invariant under the combination of $\mathbb{Z}_2\times\mathbb{Z}_2$ and inversion. This reminds me of several references discussing trivial phases protected by point group symmetries: Y. Fuji, F. Pollmann, and M. Oshikawa, Phys. Rev. Lett. \textbf{114}, 177204 (2015); A. Kshetrimayum, H-H. Tu, and R. Orus, Phys. Rev. B \textbf{93}, 245112 (2016). I wonder if it is possible to think of the ground state phase of the
$q$AKLT as a symmetry-protected trivial phase.
Response: We believe that the potential connection to symmetry-protected trivial phases is interesting and thank the reviewer for pointing it out. It may be interesting to investigate the mixing of $\pi$-rotation and bond-centered inversion symmetry in the $q$AKLT model, but we hold the view that such an investigation goes beyond the scope of this work.
-Hamiltonians with $\mathbb{Z}_2\times\mathbb{Z}_2^{(q)}$ symmetry}-
Comment: This question is somewhat related to the first: What is the most general form of a Hamiltonian that is invariant under $\mathbb{Z}_2\times\mathbb{Z}_2^{(q)}$? I know one of the authors discussed the
q-deformed bilinear-biquadratic Hamiltonian in Ref. [31], which includes the
qAKLT model as a limiting case. But does that Hamiltonian represent the most general form?
Response: We see no natural choice for the general form of a $\mathbb{Z}_2\times\mathbb{Z}_2^{(q)}$ symmetric Hamiltonian as this symmetry involves the free parameter $q$. The least artificial of these choices is to consider models with $\mathbb{Z}_2\times\mathbb{Z}_2^{(q)}$ arising from the presence of $U_q[su(2)]$ symmetry, the inevitability of which is discussed in Appendix B. The most general Hamiltonian for such a spin chain would be expressed as a polynomial in the quadratic Casimir and coproducts thereof. Indeed, the bilinear-biquadratic chain of [Quella; 2021] is the most general form for such a Hamiltonian with nearest-neighbor interactions. We have added a comment on page 8 mentioning the difficulty in choosing $\mathbb{Z}_2\times\mathbb{Z}_2^{(q)}$ symmetric couplings in the context of applying bond algebras.
-The parameter range of $q$-
Comment: In Appendix A, it is stated that a generic $q$ refers to $q>0$ and finite. However, in the main text, the parameter range of $q$. This could be problematic, as the term $C_{j,j+1}$ in Eq. (35) and its counterpart in Eq. (46) may not be positive semi-definite for general $q\in\mathbb{C}$. For instance, it is positive-semidefinite but not positive definite at $q=\pm i$ I suggest the authors explicitly specify the range of $q$ in the main text.
Response: We have added comments specifying the parameter range to be $q>0$ on pages 10 and 13, where the Witten's conjugation argument requires it.
-Explicit form of $\mathcal{N}_{q\text{KT}}$}-
Comment: Is it easy to derive the action of $\mathcal{N}_{q\text{KT}} = MU_\text{KT}M^{-1}$ on the standard basis states $\vert m_1,m_2,\dots,m_L\rangle$? It seems that the answer is almost provided in Eq. (55). However, it would still be beneficial if the authors could elaborate on this further.
Response: The action on arbitrary basis states is in general quite complicated due to the presence of the highly non-local operator $U_\text{KT}$. It is the fact that this transformation is a map between between string-ordered and ferromagnetically-ordered states that is most important here, as remarked upon at the beginning of Section 4.2. We omit the action of $\mathcal{N}_{q\text{KT}}$ on arbitrary basis states as we believe these complicated expressions are not required to understand our result.
-The existence of a spectral gap in the dual Hamiltonian $H_{q\text{SSB}}$}-
Comment: The authors argued that Knabe's method is not applicable to $H_{q\text{SSB}}$ due to its lack of translation invariance. Is this assertion really true? As discussed by Lemm and Mozgunov [J. Math. Phys. \textbf{60}, 051901 (2019)], and also in Appendix B of Ref. [32], Knabe's method has been generalized to cover systems with open boundaries. While they assume translation invariance in the bulk for the argument, this is not strictly necessary; the local interactions may depend on positions as long as the resulting Hamiltonian remains frustration-free. The situation is particularly simple when considering a subsystem of three sites to get a lower bound on a gap, in which case the subsystem Hamiltonian is defined as $h_{3,j} = H_{j,j+1}^{q\text{SSB}} + H_{j+1,j+2}^{q\text{SSB}}$. I am not sure whether one can diagonalize $h_{3,j}$ analytically. But if one can and can get a lower bound uniform in the system size $L$, one can establish the existence of a spectral gap in $H_{q\text{SSB}}$, at least for certain values of $q$.
Response: Knabe's orignal argument as well as the extensions in [Lemm-Mozgunov; 2019] and [Wouters-Katsura-Schuricht; 2021] both require that the spectral gap of the subsystem depend only on the length of the subsystem, and they do not allow for it to depend on the position of the subsystem. Although that does not mean translation-invariance is strictly required, having translation-invariance implies this property (c.f. the third comment in Section 3 of [Lemm-Mozgunov; 2019]). That is, the gap of a subsystem made from the bulk Hamiltonians $H^{q\text{SSB}}_{j,j+1}$ may depend on the position of the subsytem. The tail Hamiltonians however are translation-invariant and thus have this property. It is potentially possible to modify Knabe's method to work with the subsystem gap depending on the location of the subsystem, but our view is that that is beyond the scope of this work. We have added a comment on page 23 clarifying why Knabe's method cannot be directly applied.
-Minor comment-
Comment: I suggest the authors define $A_j^\pm$ and $A_j^0$ precisely around Eq. (2).
Response: We have added the definition and expressions for $A_j^\pm$ and $A_j^0$ in Eq.s (3) and (4) on page 4.
--Responses to referee report 2--
-Non-locality and systematic approaches}-
Comment: At the top of page 8, it is stated that ``... the non-locality means that any attempt at a correspondence fails to fit into systematic approaches to constructing dualities ...". This paragraph is a bit misleading. In the general approaches cited in this paragraph, it is made very clear that dualities are defined with respect to a certain symmetry, and that only symmetric operators remain local under the duality transformation. The usual Kennedy-Tasaki transformation is defined with respect to an ordinary Z2 x Z2 symmetry, and as the authors point out, the q-deformed AKLT Hamiltonian breaks the Z2 x Z2 symmetry.
Response: We agree with the referee that in the approaches mentioned symmetry plays a key role in constructing dualities and is directly connected to the preservation/destruction of locality and order. We have added a comment on page 8 stating explicitly that dualities are defined with respect to the global symmetry present in the approaches referenced.
-Connection with the categorical framework}-
Comment: As pointed out in the conclusion, the correct way to apply these general frameworks would be to properly identify the symmetry, which in this case is the quantum group symmetry, and dualize with respect to that symmetry. In fact, in reference [], the authors consider a duality based on the quantum group symmetry of the XXZ model, and show one can understand the well known IRF/Vertex correspondence in terms of their general categorical framework. It would be interesting to consider whether the KT transformation can be generalized by considering the $SO(3)_q$ symmetry from this general framework.
Response: The approach in [Lootens-Delcamp-Ortiz-Verstraete; 2022] does appear to be the most readily applicable, however in our context it falls short as the global symmetry one wishes to dualize with respect to must form a fusion category. Ignoring the finiteness condition for a fusion category, the authors exploit $\text{Rep}(U_q[su(2)])$ symmetry to rederive the IRF/vertex correspondence. It appears that the fact that $\text{Rep}(U_q[su(2)])$ is a semisimple tensor category is good enough to produce an accurate result (in particular the regular module category and $\text{Vec}$ as a module category are still available for $\text{Rep}(U_q[su(2)])$). In our case the global symmetry of the $q$AKLT model is $U_q[su(2)]$ itself, not its category of modules, and it cannot be framed as a tensor category. We added a comment regarding this on page 18.
-Can the operation $q\to 1/q$ be realised as a linear operator on the Hilbert space?-
Response: The operation $q\to 1/q$ itself is not implemented by a linear operator on the Hilbert space. We thought of this operation in terms of replacing $q$ with $q^{-1}$ throughout this work. It does come as a byproduct of certain linear operations, $\pi$-rotations about the $x$- and $y$-axes being the most relevant example in our work. We added a comment on page 8 explaining why it is important for us to distinguish between these two actions. The discussion in Appendix $B$ also explains how this action can be implemented on the Casimir using the universal $R$-matrix.
-Notation-
Comment: In (12), it would be a bit clearer to just write out $\vert + \rangle$ and $\vert - \rangle$, which I am assuming are defined as the eigenvectors of the Pauli X matrix, to avoid any ambiguity.
Response: On page 4 we have specified that $\{\vert \pm \rangle,\vert 0 \rangle\}$ are the standard spin-1 basis states labelled by their $S^z$ eigenvalue. This is the notation that we use throughout the manuscript.
Before responding to the referee comments individually we respond to the queries from the second reviewer regarding general applicability that were also raised by the editor: The $q$AKLT model is indeed finely-tuned so that one may construct the ground state as a matrix product state, however the physical properties, in particular string order, are part of a broader phase diagram [T. Quella Phys. Rev. B 103, 054404, 2021]. We adopt the perspective that $H_{q\text{AKLT}}$ and $H_{q\text{SSB}}$ serve as representatives of $q$-deformed SPT and SSB phases respectively. Our transformation cannot be used as a duality transformation for the entire $q$-deformed Haldane phase, however $H_{q\text{SSB}}$ indeed portrays the patterns of spontaneous symmetry breaking that we expect to hold in the dual phase. We have clarified this view in the introduction (page 3) and in the final paragraph on page 8, as well as added a further comment to this effect on pages 17-18.
Our responses to the individual comments from each referees are written below, and the changes listed appear in red in the updated manuscript.
--Responses to referee report 1--
-Possibility of a symmetry-protected trivial phase-
Comment: It is briefly mentioned in the Conclusion that the exchange of $q$ and $q^{-1}$ corresponds to the action of inversion transformation on the $q$AKLT Hamiltonian. An immediate consequence is that this Hamiltonian is invariant under the combination of $\mathbb{Z}_2\times\mathbb{Z}_2$ and inversion. This reminds me of several references discussing trivial phases protected by point group symmetries: Y. Fuji, F. Pollmann, and M. Oshikawa, Phys. Rev. Lett. \textbf{114}, 177204 (2015); A. Kshetrimayum, H-H. Tu, and R. Orus, Phys. Rev. B \textbf{93}, 245112 (2016). I wonder if it is possible to think of the ground state phase of the
$q$AKLT as a symmetry-protected trivial phase.
Response: We believe that the potential connection to symmetry-protected trivial phases is interesting and thank the reviewer for pointing it out. It may be interesting to investigate the mixing of $\pi$-rotation and bond-centered inversion symmetry in the $q$AKLT model, but we hold the view that such an investigation goes beyond the scope of this work.
-Hamiltonians with $\mathbb{Z}_2\times\mathbb{Z}_2^{(q)}$ symmetry}-
Comment: This question is somewhat related to the first: What is the most general form of a Hamiltonian that is invariant under $\mathbb{Z}_2\times\mathbb{Z}_2^{(q)}$? I know one of the authors discussed the
q-deformed bilinear-biquadratic Hamiltonian in Ref. [31], which includes the
qAKLT model as a limiting case. But does that Hamiltonian represent the most general form?
Response: We see no natural choice for the general form of a $\mathbb{Z}_2\times\mathbb{Z}_2^{(q)}$ symmetric Hamiltonian as this symmetry involves the free parameter $q$. The least artificial of these choices is to consider models with $\mathbb{Z}_2\times\mathbb{Z}_2^{(q)}$ arising from the presence of $U_q[su(2)]$ symmetry, the inevitability of which is discussed in Appendix B. The most general Hamiltonian for such a spin chain would be expressed as a polynomial in the quadratic Casimir and coproducts thereof. Indeed, the bilinear-biquadratic chain of [Quella; 2021] is the most general form for such a Hamiltonian with nearest-neighbor interactions. We have added a comment on page 8 mentioning the difficulty in choosing $\mathbb{Z}_2\times\mathbb{Z}_2^{(q)}$ symmetric couplings in the context of applying bond algebras.
-The parameter range of $q$-
Comment: In Appendix A, it is stated that a generic $q$ refers to $q>0$ and finite. However, in the main text, the parameter range of $q$. This could be problematic, as the term $C_{j,j+1}$ in Eq. (35) and its counterpart in Eq. (46) may not be positive semi-definite for general $q\in\mathbb{C}$. For instance, it is positive-semidefinite but not positive definite at $q=\pm i$ I suggest the authors explicitly specify the range of $q$ in the main text.
Response: We have added comments specifying the parameter range to be $q>0$ on pages 10 and 13, where the Witten's conjugation argument requires it.
-Explicit form of $\mathcal{N}_{q\text{KT}}$}-
Comment: Is it easy to derive the action of $\mathcal{N}_{q\text{KT}} = MU_\text{KT}M^{-1}$ on the standard basis states $\vert m_1,m_2,\dots,m_L\rangle$? It seems that the answer is almost provided in Eq. (55). However, it would still be beneficial if the authors could elaborate on this further.
Response: The action on arbitrary basis states is in general quite complicated due to the presence of the highly non-local operator $U_\text{KT}$. It is the fact that this transformation is a map between between string-ordered and ferromagnetically-ordered states that is most important here, as remarked upon at the beginning of Section 4.2. We omit the action of $\mathcal{N}_{q\text{KT}}$ on arbitrary basis states as we believe these complicated expressions are not required to understand our result.
-The existence of a spectral gap in the dual Hamiltonian $H_{q\text{SSB}}$}-
Comment: The authors argued that Knabe's method is not applicable to $H_{q\text{SSB}}$ due to its lack of translation invariance. Is this assertion really true? As discussed by Lemm and Mozgunov [J. Math. Phys. \textbf{60}, 051901 (2019)], and also in Appendix B of Ref. [32], Knabe's method has been generalized to cover systems with open boundaries. While they assume translation invariance in the bulk for the argument, this is not strictly necessary; the local interactions may depend on positions as long as the resulting Hamiltonian remains frustration-free. The situation is particularly simple when considering a subsystem of three sites to get a lower bound on a gap, in which case the subsystem Hamiltonian is defined as $h_{3,j} = H_{j,j+1}^{q\text{SSB}} + H_{j+1,j+2}^{q\text{SSB}}$. I am not sure whether one can diagonalize $h_{3,j}$ analytically. But if one can and can get a lower bound uniform in the system size $L$, one can establish the existence of a spectral gap in $H_{q\text{SSB}}$, at least for certain values of $q$.
Response: Knabe's orignal argument as well as the extensions in [Lemm-Mozgunov; 2019] and [Wouters-Katsura-Schuricht; 2021] both require that the spectral gap of the subsystem depend only on the length of the subsystem, and they do not allow for it to depend on the position of the subsystem. Although that does not mean translation-invariance is strictly required, having translation-invariance implies this property (c.f. the third comment in Section 3 of [Lemm-Mozgunov; 2019]). That is, the gap of a subsystem made from the bulk Hamiltonians $H^{q\text{SSB}}_{j,j+1}$ may depend on the position of the subsytem. The tail Hamiltonians however are translation-invariant and thus have this property. It is potentially possible to modify Knabe's method to work with the subsystem gap depending on the location of the subsystem, but our view is that that is beyond the scope of this work. We have added a comment on page 23 clarifying why Knabe's method cannot be directly applied.
-Minor comment-
Comment: I suggest the authors define $A_j^\pm$ and $A_j^0$ precisely around Eq. (2).
Response: We have added the definition and expressions for $A_j^\pm$ and $A_j^0$ in Eq.s (3) and (4) on page 4.
--Responses to referee report 2--
-Non-locality and systematic approaches}-
Comment: At the top of page 8, it is stated that ``... the non-locality means that any attempt at a correspondence fails to fit into systematic approaches to constructing dualities ...". This paragraph is a bit misleading. In the general approaches cited in this paragraph, it is made very clear that dualities are defined with respect to a certain symmetry, and that only symmetric operators remain local under the duality transformation. The usual Kennedy-Tasaki transformation is defined with respect to an ordinary Z2 x Z2 symmetry, and as the authors point out, the q-deformed AKLT Hamiltonian breaks the Z2 x Z2 symmetry.
Response: We agree with the referee that in the approaches mentioned symmetry plays a key role in constructing dualities and is directly connected to the preservation/destruction of locality and order. We have added a comment on page 8 stating explicitly that dualities are defined with respect to the global symmetry present in the approaches referenced.
-Connection with the categorical framework}-
Comment: As pointed out in the conclusion, the correct way to apply these general frameworks would be to properly identify the symmetry, which in this case is the quantum group symmetry, and dualize with respect to that symmetry. In fact, in reference [], the authors consider a duality based on the quantum group symmetry of the XXZ model, and show one can understand the well known IRF/Vertex correspondence in terms of their general categorical framework. It would be interesting to consider whether the KT transformation can be generalized by considering the $SO(3)_q$ symmetry from this general framework.
Response: The approach in [Lootens-Delcamp-Ortiz-Verstraete; 2022] does appear to be the most readily applicable, however in our context it falls short as the global symmetry one wishes to dualize with respect to must form a fusion category. Ignoring the finiteness condition for a fusion category, the authors exploit $\text{Rep}(U_q[su(2)])$ symmetry to rederive the IRF/vertex correspondence. It appears that the fact that $\text{Rep}(U_q[su(2)])$ is a semisimple tensor category is good enough to produce an accurate result (in particular the regular module category and $\text{Vec}$ as a module category are still available for $\text{Rep}(U_q[su(2)])$). In our case the global symmetry of the $q$AKLT model is $U_q[su(2)]$ itself, not its category of modules, and it cannot be framed as a tensor category. We added a comment regarding this on page 18.
-Can the operation $q\to 1/q$ be realised as a linear operator on the Hilbert space?-
Response: The operation $q\to 1/q$ itself is not implemented by a linear operator on the Hilbert space. We thought of this operation in terms of replacing $q$ with $q^{-1}$ throughout this work. It does come as a byproduct of certain linear operations, $\pi$-rotations about the $x$- and $y$-axes being the most relevant example in our work. We added a comment on page 8 explaining why it is important for us to distinguish between these two actions. The discussion in Appendix $B$ also explains how this action can be implemented on the Casimir using the universal $R$-matrix.
-Notation-
Comment: In (12), it would be a bit clearer to just write out $\vert + \rangle$ and $\vert - \rangle$, which I am assuming are defined as the eigenvectors of the Pauli X matrix, to avoid any ambiguity.
Response: On page 4 we have specified that $\{\vert \pm \rangle,\vert 0 \rangle\}$ are the standard spin-1 basis states labelled by their $S^z$ eigenvalue. This is the notation that we use throughout the manuscript.

---

## Round 1 · List of Changes

p.3: Comment specifying that we think of$H_{q\text{SSB}}$ is a representitive of an spontaneous symmetry breaking (SSB) phase.
p.4: Clarifications to notation (spin vectors and MPS tensors).
p.8(1): Clarifying the role of symmetry in systematic approaches referenced.
p.8(2): Clarifying that we do not think of $q\to q^{-1}$ as a linear map on the Hilbert space.
p.8(3)/p.9: Viewing $H_{q\text{SSB}}$ as a representitive of a $\mathbb{Z}_2\times\mathbb{Z}_2^{(q)}}$ SSB phase.
p.10, p.13: Specifying $q>0$.
p.17/p.18(1): Discussing how $H_{q\text{SSB}}$ is a representitive of an SSB phase.
p.18(2): Comments regarding obstructions to using systematic approaches, specifically addressing why the framework developed in [Lootens-Delcamp-Ortiz-Verstraete; 2022] cannot be used.
p.23: Comment on why Knabe's method cannot be directly applied.
p.4: Clarifications to notation (spin vectors and MPS tensors).
p.8(1): Clarifying the role of symmetry in systematic approaches referenced.
p.8(2): Clarifying that we do not think of $q\to q^{-1}$ as a linear map on the Hilbert space.
p.8(3)/p.9: Viewing $H_{q\text{SSB}}$ as a representitive of a $\mathbb{Z}_2\times\mathbb{Z}_2^{(q)}}$ SSB phase.
p.10, p.13: Specifying $q>0$.
p.17/p.18(1): Discussing how $H_{q\text{SSB}}$ is a representitive of an SSB phase.
p.18(2): Comments regarding obstructions to using systematic approaches, specifically addressing why the framework developed in [Lootens-Delcamp-Ortiz-Verstraete; 2022] cannot be used.
p.23: Comment on why Knabe's method cannot be directly applied.

---

## Editorial Decision

published